# Class Distribution-induced Attention Map for Open-vocabulary Semantic Segmentations

**Dong Un Kang[1], Hayeon Kim[1], Se Young Chun[1,2,†]**
[1]Department of ECE, [2]INMC & IPAI, Seoul National University
{qkrtnskfk23, khy5630, sychun}@snu.ac.kr

### Abstract

Open-vocabulary semantic segmentation is a challenging task that assigns seen or unseen class labels to individual pixels. While recent works with vision-language models (VLMs) have shown promising results in zero-shot semantic segmentation, they still struggle to accurately localize class-related objects. In this work, we argue that CLIP-based prior works yield patch-wise *noisy class predictions* while having *highly correlated class distributions* for each object. Then, we propose Class Distribution-induced Attention Map, dubbed CDAM, that is generated by the Jensen-Shannon divergence between class distributions of two patches that belong to the same (class) object. This CDAM can be used for open-vocabulary semantic segmentation by integrating it into the final layer of CLIP to enhance the capability to accurately localize desired classes. Our class distribution-induced attention scheme can easily work with multi-scale image patches as well as augmented text prompts for further enhancing attention maps. By exploiting class distribution, we also propose robust entropy-based background thresholding for the inference of semantic segmentation. Interestingly, the core idea of our proposed method does not conflict with other prior arts in zero-shot semantic segmentation, thus can be synergetically used together, yielding substantial improvements in performance across popular semantic segmentation benchmarks. Code is available at https://janeyeon.github.io/cdamclip.

## 1 Introduction

Open-vocabulary semantic segmentation aims to assign correct semantic labels in an open set of classes to each pixel of a given image. Classical semantic segmentation that assigns labels in a closed set of pre-defined classes and is trained in a supervised manner has achieved remarkable progress (Long et al., 2015; Noh et al., 2015; Chen et al., 2017; 2018; Xie et al., 2021; Yuan et al., 2020; Zhao et al., 2017b). However, the limited number of classes and laborious pixel-level human annotation have restricted the model's ability to recognize numerous seen and unseen classes in real-world settings. Open-vocabulary semantic segmentation is emerging as a promising approach for real-world applications since it allows the segmentation model to assign novel class labels at inference (Ghiasi et al., 2022; Ding et al., 2022; Xian et al., 2019; Bucher et al., 2019; Gu et al., 2020; Li et al., 2021; Liu et al., 2022; Zhao et al., 2017a; Xu et al., 2022b; Zhou et al., 2022; Cha et al., 2023; Xu et al., 2022a; Luo et al., 2023; Shin et al., 2022; Ren et al., 2023; Ranasinghe et al., 2023).

Recent advances of vision-language models (VLMs) such as ALIGN (Jia et al., 2021) and CLIP (Radford et al., 2021) have shed light on the problem of zero-shot open-vocabulary semantic segmentation for novel classes. Prior arts aimed to enhance the localization capabilities of pre-trained CLIP models for achieving great performance by 1) enhancing local alignment of VLMs between region visual and textual features with contrastive learning (Xu et al., 2022a; Luo et al., 2023; Cha et al., 2023), 2) modifying the last attention layer of CLIP without retraining (Zhou et al., 2022), or 3) leveraging self-self attention mechanisms such as query-query or key-key feature interactions within the attention map (Li et al., 2023; Bousselham et al., 2024; Wang et al., 2023; Lan et al., 2024). However, despite these advancements, prior arts still struggle with accurately localizing target objects within images.

---

†Corresponding author.

In this work, we argue that CLIP-based prior works yield patch-wise *noisy class predictions* while having *highly correlated class distributions* for each object. Then, based on these observations, we propose Class Distribution-induced Attention Map, dubbed CDAM, that is generated by the Jensen-Shannon (JS) divergence between class distributions of two patches that belong to the same (class) object for the last attention layer of CLIP. Specifically, starting from the noisy class predictions, we measure the similarity of class distributions between each patch and all other patches in the image. Since patches belonging to the same object class exhibit highly correlated class distributions, these similarities are high for same class patches and low for different class patches. By exploiting this property, these similarity scores are used to construct an attention map that refines the attention mechanism in CLIP's final layer, ensuring that attention is focused more effectively on relevant regions. Since attention map implies the significance of relevant features across different patches in the self-attention mechanism of vision transformer (ViT) (Dosovitskiy et al., 2020), our CDAM assigns high attention weights to the patches belonging to the same object class while allocating low attention weights to the patches from different object classes. This CDAM can be used for zero-shot semantic segmentation by integrating the semantic information of class-relevant patches into the final layer of CLIP to enhance the capability to accurately localize desired classes without requiring additional training or dense annotations.

Moreover, our CDAM can easily work with multi-scale image patches, augmented text prompts and entropy-based background thresholding for further enhancing semantic segmentation. CDAM with multi-scale image patches generates multiple CDAMs at various spatial scales and merges them to achieve improved spatial consistency of attention maps. CDAM with augmented text prompts such as attribute classes of common objects (*e.g.*, color and super-category) can strengthen the class distribution similarity between patches belonging to the same target class by leveraging a wider range of features to expand text class categories for enhancing attention maps. Lastly, by exploiting class distribution, we propose robust entropy-based background thresholding technique to effectively extract foreground classes from background for the inference of semantic segmentation.

Interestingly, the core idea of our proposed method does not conflict with other prior arts in zero-shot semantic segmentation and thus can be synergetically and seamlessly integrated into them for further enhanced performance. Our proposed CDAM substantially outperformed prior arts in zero-shot average mIoU (mean Intersection-over-Union) across several widely used benchmarks. The contributions of our work are summarized as:

- Proposing class distribution-induced attention map (CDAM) that yields higher weights to class-relevant patches to enhance localization capability by exploiting *robust class distribution* over *noisy class prediction* for the patches of each object class.
- Proposing CDAM with multi-scale image patches, augmented text prompts, and entropy-based background thresholding for further improving the CDAM.
- Demonstrating that our CDAM remarkably outperformed prior arts on CLIP-based training-free zero-shot semantic segmentation over diverse benchmark datasets.

## 2 RELATED WORKS

### 2.1 OPEN-VOCABULARY SEMANTIC SEGMENTATION WITH VISION-LANGUAGE MODEL

Due to the recent success of large-scale vision-language models, researchers have shown a growing interest in open-vocabulary semantic segmentation. Prior works can be broadly categorized into two approaches: First approaches leverage annotated datasets containing examples of the seen classes (Ghiasi et al., 2022; Xian et al., 2019; Bucher et al., 2019; Gu et al., 2020; Li et al., 2021; Liu et al., 2022; Xu et al., 2022b; 2023b; Jiao et al., 2023; Ge et al., 2025), and evaluates performance using mIoU for both seen and unseen classes. While some recent works (Xu et al., 2023b; Jiao et al., 2023; Ge et al., 2025) also utilize frozen CLIP features, they differ from our approach as they rely on supervised training. Specifically, they train additional mask proposal networks for object segmentation and to improve the classification of generated mask proposals. Second approaches attempt segmentation without any class-specific annotations and measures mIoU exclusively for unseen classes. These approaches train the vision-language models with image-text paired datasets based on weak supervision. GroupViT (Xu et al., 2022a) constructs the hierarchical grouping structure of transformer for localing the image regions. TCL (Cha et al., 2023) addresses suffering the train-test

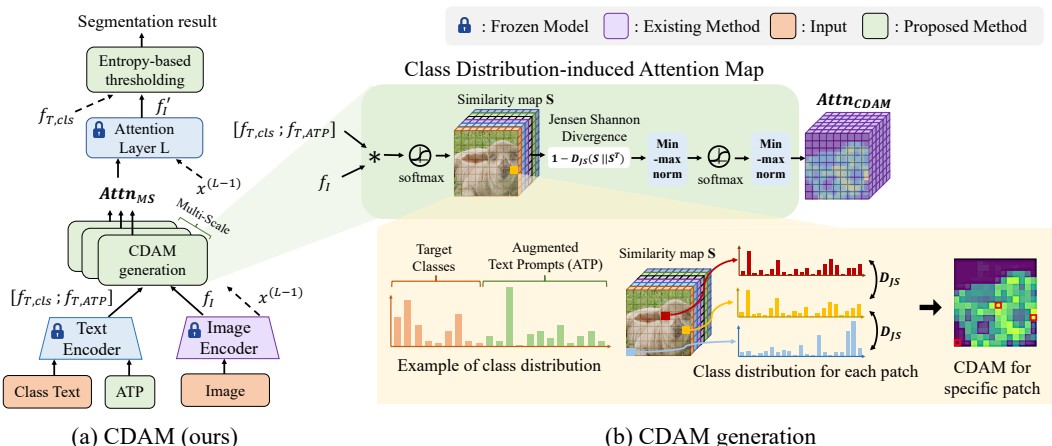

Figure 1: **The overall pipeline of our proposed CDAM.** During inference, the class distribution-induced attention map (CDAM) is constructed by measuring the distance between the class distributions of each patch in the initial similarity map $S$. The CDAM is then integrated with the last attention layer of CLIP, highlighting the class-specific regions in the input image. CDAM with multi-scale image patches and augmented text prompts can further enhance the quality of attention map. Next, we dynamically adjust the threshold value for foreground-background regions based on the entropy.

discrepancy for region-text alignment through training grounded mask decoder. SegCLIP (Luo et al., 2023) proposed semantic group module with several weakly-supervised losses. Recent training-free methods, like CLIPSurgery (Li et al., 2023), SCLIP (Wang et al., 2023) and GEM (Bousselham et al., 2024), use self-self attention mechanisms, specifically query-query, key-key or value-value attention, to capture similar characteristic patches in the attention map. In contrast, our CDAM generates an attention map based on image-text feature similarity from class distribution. Consequently, our CDAM originates from the initial noisy predictions of existing methods, and our approach can be easily integrated into other training-free methods to enhance their localization capabilities.

## 2.2 BACKGROUND SUBTRACTION

In image processing, separating foreground objects from the background is crucial for many applications. Thresholding provides a simple and effective technique to achieve this by classifying pixels with intensity values below a chosen threshold as background and those above as foreground. In grayscale images, various thresholding approaches are categorized based on the type of information they use, including: histogram shape information (Rosenfeld & De La Torre, 1983), measurement space clustering (Otsu et al., 1975; Sezan, 1990; Olivo, 1994), histogram entropy information (Li & Lee, 1993; Pal, 1996), image attribute information (Tsai, 1985), spatial information (Pal & Pal, 1989), and local characteristics (Sauvola & Pietikäinen, 2000). However, in open-vocabulary semantic segmentation, the background class is considered "unknown" and distinct from the foreground classes with specific labels. This makes background subtraction more challenging compared to traditional image processing methods. In this paper, we dynamically adjust the threshold value for discrimination of the background region considering the entropy of the class distribution.

## 3 METHODS

The key contribution of our proposed method is to enhance the localization ability of large-scale vision-language models for open-vocabulary semantic segmentation without additional training and annotations. Starting with an initial, potentially inaccurate prediction, we introduce the Class Distribution-induced Attention Map (CDAM), an approach that emphasizes the attention weight on patch regions relevant to specific classes within the attention map of last attention layer. Additionally, we propose a entropy-based background thresholding technique that adaptively distinguishes between foreground and background regions. The overall pipeline of our CDAM is illustrated in Fig. 1.

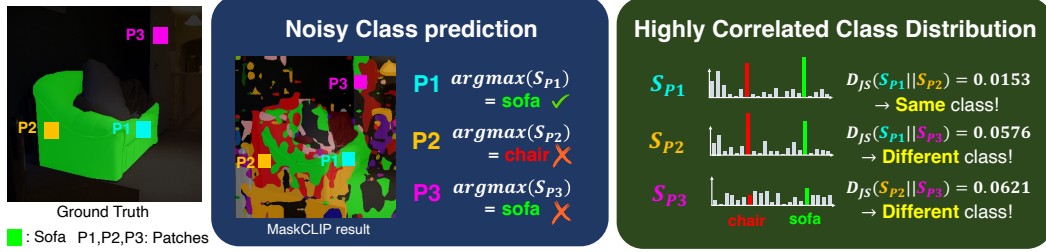

Figure 2: **Similarity of class distributions between patches.** From the noisy prediction of MaskCLIP (Zhou et al., 2022), we explore the similarity of class distributions between patches. $S_{Pi}$ represents the class distribution at the position of patch $Pi$ within the similarity map $S$. Although the segmented classes differ, the similarity of the class distribution of patches between true positive ($P1$) and false negative ($P2$) is more similar than between true positive ($P1$) and false positive ($P3$). The distance of class distribution is measured by JS divergence, $D_{JS}(p||q)$.

## 3.1 LIMITATION OF SEMANTIC SEGMENTATION WITH VISION-LANGUAGE MODEL

CLIP (Radford et al., 2021) is a pre-trained VLM with 400 million curated image-text paired dataset. In the self-attention mechanism of its transformer-based image encoder, the attention map reflects the relationships among visual tokens. The attention weight is computed using the similarity between pairs of query and key embeddings within each attention layer. Taking the flatten feature map $\boldsymbol{x} \in \mathbb{R}^{N \times D}$ where $N$ denotes the number of tokens and $D$ refers to the embedding dimension, the attention map is formulated as

$$\mathbf{Attn}(\boldsymbol{Q}, \boldsymbol{K}) = \text{Softmax}(\boldsymbol{Q}\boldsymbol{K}^T/\sqrt{D}) \in \mathbb{R}^{N \times N} \tag{1}$$

where the query and key embeddings, $\boldsymbol{Q} = \boldsymbol{x}\boldsymbol{W}_q$ and $\boldsymbol{K} = \boldsymbol{x}\boldsymbol{W}_k$, are obtained using the projection matrices $\boldsymbol{W}_q$ and $\boldsymbol{W}_k \in \mathbb{R}^{D \times D}$, respectively. Thus, the output of the self-attention will be $\mathbf{Attn}(\boldsymbol{Q}, \boldsymbol{K})\boldsymbol{V}$ where $\boldsymbol{V} = \boldsymbol{x}\boldsymbol{W}_v$ and $\boldsymbol{W}_v \in \mathbb{R}^{D \times D}$. The attention map, the output of $\mathbf{Attn}(\cdot, \cdot)$, is crucial for capturing long-range semantic dependencies between patch tokens in image recognition. However, due to image-level texts for pre-training, CLIP is not usually applicable for dense prediction tasks like semantic segmentation where precise localization of target classes is essential.

To mitigate the limitation of CLIP, the attention map should be adjusted to weigh more on class-relevant region for each patch token. MaskCLIP (Zhou et al., 2022) extends the pre-trained CLIP model to perform dense predictions by minimally modifying the last layer of the image encoder. Specifically, the original attention map is replaced by the identity matrix $\boldsymbol{I} \in \mathbb{R}^{N \times N}$, removing the query and key embedding layers in the self-attention of the last layer and thus the output of the self-attention in the last layer is $\boldsymbol{V}$. However, using the identity matrix as the last attention map may overly emphasize the patch embedding itself and neglects information from class-relevant neighboring patches, thus leading to inaccurate segmentation quality.

## 3.2 CLASS DISTRIBUTION-INDUCED ATTENTION MAP

Our approach aims to enhance the quality of the attention map of the last layer of CLIP in Eq. (1) for CLIP-based open-vocabulary semantic segmentation methods.

### 3.2.1 HIGHLY CORRELATED CLASS DISTRIBUTIONS VS. NOISY CLASS PREDICTIONS

The class distribution learned by the pre-trained model includes rich information about recognition patterns (Hinton et al., 2015). We conjecture that the pre-trained CLIP model implicitly captures knowledge about target classes through its class distribution. We chose a prior art on CLIP based semantic segmentation, MaskCLIP (Zhou et al., 2022), and carefully observed the output of it for a toy example with 'sofa' by selecting two patches ($P1$, $P2$) that belong to it and one patch ($P3$) that does not as illustrated in Fig. 2. MaskCLIP yielded locally noisy 'sofa' output with the maximum likelihood class prediction per patch ($P2$, $P3$), but surprisingly also yielded highly correlated class distributions between the same class patches ($P1$, $P2$) and somewhat uncorrelated across the different class patches ($P1$, $P3$), thus confirming our conjecture. To support these observation, we use several CLIP-based training-free methods as baseline model and extend the analysis with benchmark datasets.

For a given image, one patch $P_{target}$ was randomly selected and then two patches $P_{in}$ and $P_{out}$ were randomly selected from the target class region and the rest of the region, respectively. Then, we measure (1) the probability that class prediction in $P_{target}$ is correct and (2) the probability that distribution similarity between $P_{target}$ and $P_{in}$ is higher than distribution similarity between $P_{target}$ and $P_{out}$. The results show that while CLIP-based baseline methods exhibit relatively low accuracy in class predictions for $P_{target}$ (e.g., 67.0% and 70.8% on VOC21, and 33.6% and 37.5% on COCO-Obj for SCLIP (Wang et al., 2023) and GEM (Bousselham et al., 2024), respectively), they perform significantly better in identifying class distribution similarity between patches of the same object class (e.g., 78.9% and 79.4% on VOC21, and 75.4% and 74.2% on COCO-Obj). More detailed results are provided in the supplementary materials. These similarity and dissimilarity can be measured by the Jensen-Shannon (JS) divergence and thus can be incorporated potentially for more precise attention weights for the patches belonging to the same object.

### 3.2.2 CLASS DISTRIBUTION-INDUCED ATTENTION MAP

Here we propose class distribution-induced attention map, dubbed CDAM, that utilizes the class distributions from most CLIP-based semantic segmentation methods with dense predictions. Firstly, the dense visual features and text features are extracted for the input image $z_I$ and the text prompts for each class name of the target objects $z_{T,cls}$ using the image and text encoders $\mathcal{E}_I$ and $\mathcal{E}_T$ as $\boldsymbol{f}_I = \mathcal{E}_I(z_I) \in \mathbb{R}^{(N-1)\times d}$ and $\boldsymbol{f}_{T,cls} = \mathcal{E}_T(z_{T,cls}) \in \mathbb{R}^{C\times d}$, respectively, where $C$ and $d$ refer to the number of target classes and the projected output space dimension, respectively. Then, we proposed to measure the distance between class distributions using the JS divergence $D_{JS}(\cdot||\cdot)$, a finite and symmetric metric. Min-max normalization was applied before and after softmax operation, but was omitted for simplicity. Thus, our class distribution-induced attention map is formulated as:

$$\textbf{Attn}_{\text{CDAM}} = \text{Softmax}(\{1 - D_{JS}(\mathbf{S}||\mathbf{S}^T)\}/\tau) \tag{2}$$

where $\mathbf{S} = \text{Softmax}(\rho(\boldsymbol{f}_I, \boldsymbol{f}_T)/\tau) \in \mathbb{R}^{(N-1)\times C}$ is the similarity map between dense visual features and text features, $\rho$ denotes the cosine similarity and the temperature $\tau$ controls the softness of attention such as $\sqrt{D}$ in the original self-attention layer, Eq. (1). Our CDAM was designed to weigh more on class-relevant patches with the source patch.

### 3.2.3 REFINEMENTS OF CLASS DISTRIBUTION-INDUCED ATTENTION MAP

**CDAM with multi-scale image patches.** Our class distribution conjecture is valid over different space scales. Thus, we propose the multi-scale structure of CDAMs by constructing CDAMs with downsampling-upsampling at different scales in the set of scaling factors $M$ and aggregating them as:

$$\textbf{Attn}_{\text{MS}} = \frac{1}{|M|} \sum_{m \in M} \textbf{Attn}_{\text{CDAM},m} \tag{3}$$

where $\textbf{Attn}_{\text{CDAM},m} = \text{Up}_m[\text{Softmax}(\{1 - D_{JS}(\text{Dn}_m[\mathbf{S}]||\text{Dn}_m[\mathbf{S}]^T)\}/\tau)]$ and $\text{Up}_m[\cdot]$ and $\text{Dn}_m[\cdot]$ denote the upsampling and downsampling operations at the scale $m$, respectively. This multi-scale CDAM $\textbf{Attn}_{\text{MS}}$ helped to refine CDAMs with enhanced spatial consistency.

**CDAM with augmented text prompts.** We propose to incorporate the names of attributes classes (*e.g.*, yellow, fabric, striped) and super-category (*e.g.*, animal, indoor, food) of common objects, dubbed augmented text prompts (ATP), to enrich the representation of implicit knowledge within class distribution. PACO (Ramanathan et al., 2023) provides 59 attribute classes for common objects, encompassing properties like color, pattern, material and transparency. Additionally, COCO-Stuff (Caesar et al., 2018) and MSCOCO (Lin et al., 2014) offer 12 and 15 super-categories, respectively. We filtered out the overlapping classes and removed vague terms like 'others'. Then, we used 80 augmented text prompts, $z_{T,ATP}$, and add them to the original target class text prompts. Therefore, we extract the text features of augmented text prompts $\boldsymbol{f}_T = [\boldsymbol{f}_{T,cls}; \boldsymbol{f}_{T,ATP}] = [\mathcal{E}_T(z_{T,cls}); \mathcal{E}_T(z_{T,ATP})] \in \mathbb{R}^{(C+80)\times d}$ only for constructing CDAM, not for inferencing with it. These additional classes in text prompts contribute to enhancing the similarity of class distributions among class-relevant patches.

### 3.2.4 OPEN-VOCABULARY SEMANTIC SEGMENTATION WITH CDAM

The overall inference process is visualized in Fig. 1. First, we generate the initial similarity map $\mathbf{S}$ from dense prediction of existing methods using CLIP, such as MaskCLIP (Zhou et al., 2022) and

SCLIP (Wang et al., 2023). The CLIP consists of $L$ attention layers. Second, we construct the CDAM with multi-scale image patches $\textbf{Attn}_{MS}$ by measuring the similarity of class distribution within the initial similarity map $\textbf{S}$. Finally, we incorporate this localized attention map, $\textbf{Attn}_{MS}$, into the last attention layer of CLIP to compute the final similarity map $\textbf{S}$. We reuse the latent features from the $L-1$th attention layer of CLIP, $\boldsymbol{x}^{(L-1)}$, as value features. Note that augmented text prompts are not used for computing final similarity map $\textbf{S}$. The visualized examples of class distribution-induced attention maps are shown in the supplementary materials.

### 3.3 ENTROPY-BASED BACKGROUND THRESHOLDING

For segmenting the background class that excludes all target classes, a class probability thresholding approach has been commonly used in class prediction with the probability to be 1 for foreground patches and 0 for background patches with the default threshold value $\text{Thr}_{default} = 0.5$. In real cases, however, one must adjust the threshold value by scaling $\alpha$ in the optimal threshold $\alpha\text{Thr}_{default}$ considering the uncertainty for belonging to the both foreground and background. However, it is challenging to finding an optimal $\alpha$ for the whole image. Our proposed CDAM allows us to exploit class distribution per patch by information-theoretic measures.

Here we propose an entropy-based background thresholding method that dynamically adjusts threshold values. For the similarity map $\boldsymbol{S}$, the entropy of it is $H(\boldsymbol{S}) = -\sum_{i=1}^{C} \boldsymbol{S}^i \log \boldsymbol{S}^i$ where $\boldsymbol{S}^i$ is the probability of the $i$th class. Then, we can conjecture that the foreground that belongs to an object class yields low entropy due to high confidence while the background yield high entropy due to high uncertainty. Then, we define the center entropy value $\text{H}(\textbf{S})_{center}$ that is the average of the maximum and minimum values of $H(\boldsymbol{S})$ over all image patches. Thus, our entropy-based background thresholding is formulated as

$$\text{Thr}_{\text{ent-bg}} = \alpha\text{Thr}_{default}/\text{H}(\textbf{S})_{center} \tag{4}$$

where $\text{Thr}_{\text{ent-bg}}$ represents the entropy-based background threshold value and $\alpha$ refers to a hyperparameter. Highly correlated class distribution also helps to determine a stable threshold over existing methods based on noisy class prediction. We empirically verified the effectiveness of our entropy-based background thresholding by comparing it with existing thresholding methods in image processing in the supplementary material.

## 4 EXPERIMENTS

### 4.1 EXPERIMENTAL SETUP

**Datasets.** We evaluate our CDAM method on the three widely used benchmark datasets that include a background class: PASCAL VOC (Everingham et al., 2010), PASCAL Context (Mottaghi et al., 2014) and COCO-Object (Lin et al., 2014). All three datasets include a background class, which is separate from the foreground classes. These datasets have 20, 59, and 80 foreground classes, respectively. The validation sets contain 1449, 5105, and 5000 images, respectively. We also use three additional benchmark datasets that do not include a background class: CityScapes (Cordts et al., 2016), ADE20K (Zhou et al., 2017), and COCO-Stuff (Lin et al., 2014), which have 19, 150, and 171 classes, respectively.

**Unified evaluation protocol.** We follow the unified evaluation protocol by TCL (Cha et al., 2023) in open-vocabulary semantic segmentation. This protocol ensures no access to target data before evaluation. It prohibits dataset-specific hyperparameter tuning or tricks like expanding or rephrasing class names. We fixed the background thresholding hyperparameters across datasets. To ensure a fair comparison, we reproduced existing CLIP-based training-free methods, including MaskCLIP (Zhou et al., 2022), SCLIP (Wang et al., 2023), CaR (Sun et al., 2024), GEM (Bousselham et al., 2024) and ClearCLIP (Lan et al., 2024), following the unified protocol and eliminating renaming tricks. The reproduction details are described in supplementary material.

**Implementation details.** In our CDAM model, we utilize the CLIP ViT/B-16 model from Open-CLIP (Radford et al., 2021), trained on the LAION dataset (Schuhmann et al., 2022). The input image is resized to 224 x 224 pixels, and the patch size is set to 16 x 16 pixels. Following the experimental settings of GroupViT (Xu et al., 2022a), we resize input images to have the shorter

Table 1: **Comparison with state-of-the-art methods on benchmark datasets with background class.** We evaluate the open-vocabulary semantic segmentation methods on VOC21 (Everingham et al., 2010), Context60 (Mottaghi et al., 2014) and COCO-Obj (Lin et al., 2014). SD stands for Stable Diffusion (Rombach et al., 2022) and we marked $^\dagger$ for the reproduced methods by following the unified evaluation protocol (Cha et al., 2023) and removing renaming tricks. For each dataset, we highlighted the best performance in bold and underlined the second-best performance. Performance improvements by CDAM are indicated in parentheses. The evaluation is based on mIoU (%).

| Method | Pre-trained Model | Extra Training | VOC21 | Context60 | COCO-Obj | Avg. |
|---|---|---|---|---|---|---|
| *weakly-supervised methods with additional training dataset* | | | | | | |
| GroupViT (Xu et al., 2022a) | Scratch | ✓ | 50.4 | 18.7 | 27.5 | 32.2 |
| CLIPpy (Ranasinghe et al., 2023) | CLIP | ✓ | 52.2 | - | 32.0 | - |
| ViewCo (Ren et al., 2023) | Scratch | ✓ | 52.4 | 23.0 | 23.5 | 33.0 |
| SegCLIP (Luo et al., 2023) | CLIP | ✓ | 52.6 | 24.7 | 26.5 | 34.6 |
| OVsegmentor (Xu et al., 2023a) | DINO | ✓ | 53.8 | 20.4 | 25.1 | 33.1 |
| TCL (Cha et al., 2023) | CLIP | ✓ | 51.2 | 24.3 | 30.4 | 35.3 |
| PACL (Mukhoti et al., 2023) | CLIP | ✓ | 72.3 | 50.1 | - | - |
| *visual prototype generation methods for each concept* | | | | | | |
| OVDiff (Karazija et al., 2023) | CLIP +SD+DINO | ✗ | 67.1 | 30.1 | 34.8 | 44.0 |
| FreeDA (Barsellotti et al., 2024) | CLIP +SD+DINO | ✗ | 55.4 | 38.3 | 37.4 | 43.7 |
| *CLIP-based training-free methods* | | | | | | |
| CLIPSurgery (Li et al., 2023) | CLIP | ✗ | - | 29.3 | - | - |
| CLIP-DIY (Wysoczańska et al., 2024) | CLIP+DINO | ✗ | 59.0 | - | 30.4 | - |
| CaR$^\dagger$ (Sun et al., 2024) | CLIP | ✗ | **59.4** | 25.0 | 33.2 | 39.2 |
| MaskCLIP$^\dagger$ (Zhou et al., 2022) | CLIP | ✗ | 33.1 | 23.3 | 24.8 | 27.1 |
| MaskCLIP+**CDAM** | CLIP | ✗ | 55.9 (+22.8) | 30.5 (+7.2) | 34.3 (+9.5) | 40.2 (+13.1) |
| SCLIP$^\dagger$ (Wang et al., 2023) | CLIP | ✗ | 50.5 | 25.8 | 31.3 | 35.9 |
| SCLIP+**CDAM** | CLIP | ✗ | 59.0 (+8.5) | 30.4 (+4.5) | 34.5 (+3.0) | 41.3 (+5.4) |
| ClearCLIP$^\dagger$ (Lan et al., 2024) | CLIP | ✗ | 50.7 | 27.8 | 33.0 | 37.2 |
| ClearCLIP+**CDAM** | CLIP | ✗ | 57.6 (+6.9) | 29.8 (+2.0) | 34.5 (+1.5) | 40.6 (+3.4) |
| GEM$^\dagger$ (Bousselham et al., 2024) | CLIP | ✗ | 52.1 | 28.1 | 33.8 | 38.0 |
| GEM+**CDAM** | CLIP | ✗ | 58.7 (+6.6) | **30.6** (+2.5) | **35.2** (+1.4) | **41.5** (+3.5) |

side of 448 pixels and employ the mean Intersection-over-Union (mIoU) metric, which is generally used for evaluating semantic segmentation performance. To ensure a fair comparison, Pixel-Adaptive Mask Refinement (PAMR) (Araslanov & Roth, 2020) as post-processing were not applied to any of the evaluated methods. The temperature $\tau$ and the modulation of entropy $\alpha$ are set to 0.1 and 2.5, respectively. The set of scaling factor $M$ is $\{0.25, 0.37, 0.5, 0.63, 0.75, 0.87, 1.0\}$.

## 4.2 COMPARISON WITH STATE-OF-THE-ART METHODS

**Datasets with background class.** As shown in the Table 1, we compare our CDAM with existing open-vocabulary semantic segmentation methods on VOC21, Context60 and COCO-Obj, including background class. First, weakly-supervised methods requires large-scale image-text paired training datasets such as CC12M (Changpinyo et al., 2021) and YFCC (Thomee et al., 2016). Additionally, the results of specific methods, including OVsegmentor (Xu et al., 2023a), SegCLIP (Luo et al., 2023), and ViewCo (Ren et al., 2023), were obtained by tuning dataset-specific hyperparameters for background thresholding, significantly improving the segmentation performance. In contrast, our CDAM enhances the localization ability of existing methods using a pre-trained CLIP model during inference and adaptively adjusts the thresholding value. As a result, our method, even when incorporated with MaskCLIP (Zhou et al., 2022), outperforms weakly-supervised methods on all benchmark datasets except for PACL (Mukhoti et al., 2023). Second, we compare our proposed method with training-free methods. Our CDAM effectively enhances the localization ability of existing methods, significantly achieving a performance improvement over VOC21, Context60, and COCO-Obj. Using a single model CLIP, our method with GEM (Bousselham et al., 2024) performs better than visual prototype generation methods that use three large-scale foundation models, specifically OVDiff (Karazija et al., 2023) on Context60 and COCO-Obj, and FreeDA (Barsellotti et al., 2024) on VOC21. While CaR (Sun et al., 2024) requires high computational costs and CLIP-DIY (Wysoczańska et al., 2024) employs an additional pre-trained background extractor, FOUND (Siméoni et al., 2023), we surpass

Table 2: **Comparison with state-of-the-art methods on benchmark datasets without background class.** We evaluate the open-vocabulary semantic segmentation methods on COCO-Stuff (Lin et al., 2014), CityScapes (Cordts et al., 2016) and ADE20K (Zhou et al., 2017). We marked [†] for the reproduced methods. For each dataset, we highlighted the best performance in bold and underlined the second-best performance. Performance improvements by CDAM are indicated in parentheses. The evaluation is based on mIoU (%).

| Method | Pre-trained Model | Extra Training | COCO-Stf | CityScapes | ADE20K | Avg. |
|---|---|---|---|---|---|---|
| *weakly-supervised methods with additional training dataset* | | | | | | |
| GroupViT (Xu et al., 2022a) | Scratch | ✓ | 15.3 | 11.1 | 9.2 | 11.9 |
| CLIPpy (Ranasinghe et al., 2023) | CLIP | ✓ | - | - | 13.5 | - |
| SegCLIP (Luo et al., 2023) | CLIP | ✓ | - | 11.0 | 8.7 | - |
| TCL (Cha et al., 2023) | CLIP | ✓ | 19.6 | 23.1 | 14.9 | 19.2 |
| PACL (Mukhoti et al., 2023) | CLIP | ✓ | 38.8 | - | 31.4 | - |
| *visual prototype generation methods for each concept* | | | | | | |
| FreeDA (Barsellotti et al., 2024) | CLIP +SD+DINO | ✗ | 27.8 | 36.7 | 22.4 | 29.0 |
| *CLIP-based training-free methods* | | | | | | |
| MaskCLIP[†] (Zhou et al., 2022) | CLIP | ✗ | 16.5 | 23.8 | 12.2 | 17.5 |
| MaskCLIP+**CDAM** | CLIP | ✗ | 24.5 (+8.0) | 27.6 (+3.8) | **17.8** (+5.6) | **23.3** (+5.8) |
| SCLIP[†] (Wang et al., 2023) | CLIP | ✗ | 21.1 | 19.7 | 14.6 | 18.5 |
| SCLIP+**CDAM** | CLIP | ✗ | 24.5 (+3.4) | 24.6 (+4.9) | 17.2 (+2.6) | 22.1 (+3.6) |
| ClearCLIP[†] (Lan et al., 2024) | CLIP | ✗ | 23.9 | 20.8 | 16.6 | 20.4 |
| ClearCLIP+**CDAM** | CLIP | ✗ | 24.6 (+0.7) | 21.7 (+0.9) | 17.1 (+0.5) | 21.1 (+0.7) |
| GEM[†] (Bousselham et al., 2024) | CLIP | ✗ | 23.7 | 21.2 | 15.7 | 20.2 |
| GEM+**CDAM** | CLIP | ✗ | **24.8** (+1.1) | 23.7(+1.5) | 17.2 (+1.5) | 21.9 (+1.7) |

them in averaged zero-shot performance without additional pre-trained models and with minimal computational costs. Notably, prior works such as CaR (Sun et al., 2024) or CLIP-DIY (Wysoczańska et al., 2024) are structurally incompatible with our CDAM, as they classify regions using CLS tokens, whereas CDAM relies on local visual tokens.

**Datasets without background class.** We further evaluate the effectiveness of our CDAM by incorporating it with existing methods on 3 benchmark datasets that do not include a background class. Entropy-based background thresholding, a component of CDAM, was originally designed to handle real-world scenarios with background classes. To evaluate performance without backgrounds, we disable the entropy-based background thresholding in CDAM. In Table 2, we demonstrated that our CDAM synergetically and consistently enhance the performance of existing CLIP-based training-free methods. Notably, MaskCLIP (Zhou et al., 2022) with CDAM outperformed weakly-supervised and training-free methods, except for PACL (Mukhoti et al., 2023) that requires substantially large-scale training datasets, and FreeDA (Barsellotti et al., 2024) that employs 3 large foundation models. Without the background class, evaluation focuses solely on target classes, minimizing the impact of background prediction errors. Our method reduces false positives in both target and background regions, with entropy-based thresholding improving results when the background is evaluated. However, these gains are less impactful, observed in smaller improvements in Table 2, compared to Table 1.

## 4.3 ABLATION STUDY AND ANALYSIS

To validate the importance of each component in our CDAM model, we conducted ablation studies. We employed MaskCLIP (Zhou et al., 2022), SCLIP (Wang et al., 2023), ClearCLIP (Lan et al., 2024), and GEM (Bousselham et al., 2024) as baseline models.

**Class distribution-induced attention map ($Attn_{CDAM}$ and $Attn_{MS}$).** We first investigate the effectiveness of a class distribution-induced attention map for open-vocabulary semantic segmentation. Applying our proposed $Attn_{CDAM}$ significantly improves performance across all baseline models. Notably, $Attn_{CDAM}$ achieves this improvement without any additional training and annotations. Moreover, considering the spatial consistency based on CDAM with multi-scale image patches, $Attn_{MS}$, it leads to a substantial performance boost.

Table 3: **Ablation study on components of our CDAM.** The proposed components of our CDAM consistently improve the semantic segmentation performance on VOC21 (Everingham et al., 2010) dataset. ATP refers to the augmented text prompts. The evaluation is based on mIoU(%).

| Baseline | Attn$_{CDAM}$ | Attn$_{MS}$ | ATP | Thr$_{ent-bg}$ | MaskCLIP | SCLIP | ClearCLIP | GEM |
|:---:|:---:|:---:|:---:|:---:|:---:|:---:|:---:|:---:|
| ✓ | | | | | 33.1 | 50.5 | 50.7 | 52.1 |
| ✓ | ✓ | | | | 50.1 | 55.0 | 52.1 | 54.7 |
| ✓ | ✓ | ✓ | | | 53.7 | 56.9 | 55.8 | 56.5 |
| ✓ | ✓ | ✓ | ✓ | | 54.7 | 57.2 | 56.0 | 56.9 |
| ✓ | ✓ | ✓ | ✓ | ✓ | **55.9** | **59.0** | **57.6** | **58.7** |

**Augmented text prompts (ATP).** Next, we leverage the 80 augmented text prompts to enrich the representation of the class distribution, leading to consistent performance improvement. Incorporating augmented text prompts leads to a higher similarity between the class distributions of class-relevant patches and enhances the localized class distribution-induced attention map. This suggests that our CDAM can effectively integrate the text information of the target class and additional context into the attention process.

**Entropy-based background thresholding (Thr$_{ent-bg}$).** Finally, we apply the entropy-based background thresholding which adjusts threshold based on the entropy. Without dataset-specific hyperparameter tuning, Thr$_{ent-bg}$ significantly improves the performance with several baseline methods. It suggests that even though Thr$_{ent-bg}$ is developed through the empirical experiments, it affects performance improvement over all baseline models.

To analyze (1) inference time as computation cost and (2) the metrics for measuring similarity of class distribution for our CDAM, we conducted additional experiments. The detailed results and analysis are provided in the supplementary materials.

**Inference time.** As shown in Table 5 and 6, our CDAM introduces minimal computational overhead, with an increase of at most 34 ms, which remains suitable for real-time applications. Notably, CDAM achieves approximately 200 times faster inference compared to CaR (Sun et al., 2024) on the COCO-Obj dataset. Moreover, the majority of the increased inference time is attributed to the Attn$_{MS}$ and ATP modules, primarily due to the computational burden of calculating JS divergence.

**Correlation metric.** The JS divergence is suitable for constructing our CDAM due to its symmetry and bounded range, providing better robustness in generating attention maps compared to metrics such as KL divergence (asymmetric and unbounded) and Wasserstein distance (symmetric but unbounded). Empirically, Table 8 demonstrates that JS divergence achieved the best average performance across several baseline methods. For these reasons, JS divergence was selected for our CDAM.

## 4.4 QUALITATIVE RESULTS

**Effect of CDAM components.** In Fig. 3, we qualitatively visualize examples of segmentation results by adding components of CDAM. Without relying on annotations or training, our proposed (Attn$_{CDAM}$) can effectively reduce the noisy class prediction of baseline model (MaskCLIP (Zhou et al., 2022)) and localize the target classes within an image. The quality of the segmentation results is further enhanced by two factors: (1) improved spatial consistency achieved through CDAM with multi-scale image patches, and (2) enhanced similarity of class distributions between class-relevant patches by incorporating the augmented text prompts (ATP).

**Qualitative segmentation results.** As shown in Fig. 4, we qualitatively demonstrate the improvements in localizing the target object using the proposed CDAM with Attn$_{MS}$. Compared to the initial predictions from existing CLIP-based training-free methods, integrating our approach reduces noisy predictions, resulting in smoother and more accurate outcomes. Remarkably, as we observed in Fig. 2, our method is able to generate clean and accurate attention maps even from noisy predictions. Our generated CDAM effectively highlights patches within same objects class like trains and sheep, significantly enhancing segmentation quality by making predictions more precise and less noisy. Notably, we observed that our attention map can capture highly detailed features. For example, it can distinguish elements such as the train's doors or the fence in front of the sheep in the input images.

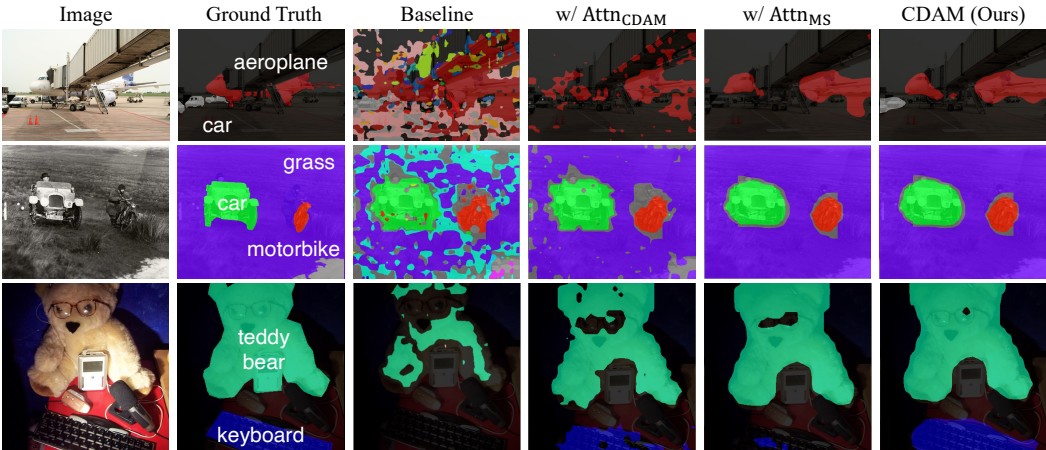

Figure 3: **Qualitative effectiveness of CDAM components.** Each component of CDAM consistently improves the quality of the semantic segmentation, leading to less noisy predictions and improved localization accuracy. We use MaskCLIP (Zhou et al., 2022) as baseline model.

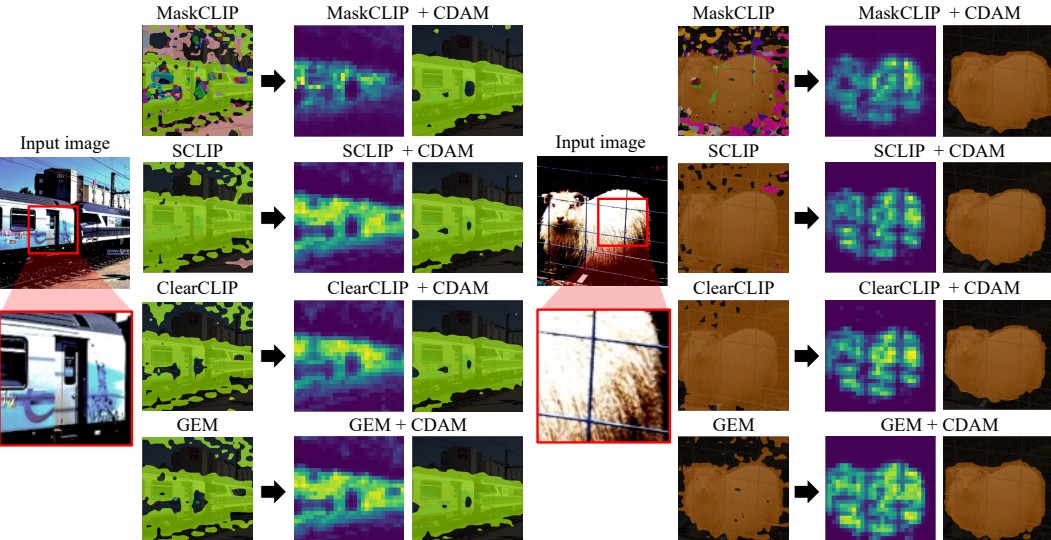

Figure 4: **Qualitative segmentation results of CDAM from inaccurate initial predictions.** Our proposed CDAM demonstrated its ability to generate high-quality attention maps (Attn$_{MS}$) even when starting from inaccurate predictions provided by prior methods. This capability led to significantly reduced noise in the final predictions of CDAM. Notably, our CDAM captures fine-grained details present within images, such as doors in a train and fence in front of sheep.

## 5 CONCLUSION

This paper proposes CDAM, a training-free approach for open-vocabulary semantic segmentation. We exploit the observation of highly correlated class distribution between class-relevant patches, even when the object within an image are inaccurately segmented. Based on this, we construct a localized, class distribution-induced attention map during inference. This map allocates higher attention weights to patches likely to belong to the same object class. Furthermore, the proposed entropy-based background thresholding adjusts the threshold value dynamically, thereby improving discrimination for foreground and background patches. CDAM can be seamlessly integrated into other training-free methods using CLIP, and we demonstrate its effectiveness through evaluations on benchmark datasets, achieving substantial improvements in semantic segmentation.

ACKNOWLEDGEMENTS

This work was supported in part by Institute of Information & communications Technology Planning & Evaluation (IITP) grant funded by the Korea government(MSIT) [NO.RS-2021-II211343, Artificial Intelligence Graduate School Program (Seoul National University)], the National Research Foundation of Korea(NRF) grant funded by the Korea government(MSIT) (No. NRF-2022M3C1A309202211), AI-Bio Research Grant through Seoul National University and Samsung Electronics Co., Ltd (IO221004-02674-01). Also, the authors acknowledged the financial support from the BK21 FOUR program of the Education and Research Program for Future ICT Pioneers, Seoul National University.

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

## A DETAILS OF AUGMENTED TEXT PROMPTS

We utilize augmented text prompts, including attribute classes and supercategories of common objects. From PACO (Ramanathan et al., 2023), we obtained the 59 attribute classes, which consist of color (31 classes), pattern (8 classes), material (16 classes), and transparency (4 classes). For general text prompts, we filtered out the four vague classes: "others(color)", "others(pattern marking)", "others(material)" and "others(transparency)". Next, from COCO-Stuff (Caesar et al., 2018) and MSCOCO (Lin et al., 2014), we obtain the 12 and 15 supercategories, respectively. We remove the overlapped class "furniture" and "food". Thus, we utilize the 80 augmented text prompts (55 attribute classes and 25 supercategories) and the list of augmented text prompts is described in Table. 4.

Table 4: **List of augmented text prompts.** We list the 80 augmented text prompts used in CDAM, including attribute classes and supercategories.

| | Source | List of class names |
|---|---|---|
| **Attribute class** | PACO | black, light blue, blue, dark blue, light brown, brown, dark brown, light green, green, dark green, light grey, grey, dark grey, light orange, orange, dark orange, light pink, pink, dark pink, light purple, purple, dark purple, light red, red, dark red, white, light yellow, yellow, dark yellow, plain, striped, dotted, checkered, woven, studded, perforated, floral, logo, text, stone, wood, rattan, fabric, crochet, wool, leather, velvet, metal, paper, plastic, glass, ceramic, opaque, translucent, transparent |
| **Supercategory** | COCO-Stuff MSCOCO | person, vehicle, outdoor, animal, accessory, sports, kitchen, food, furniture, electronic, appliance, indoor, water, ground, solid, sky, structural, building, textile, window, floor, ceiling, wall, rawmaterial, plant |

## B DETAILS OF COMPUTATIONAL COSTS

**Analysis of inference time for CDAM.** We analyzed the computational complexity of our CDAM by measuring its inference time (in seconds per image). Specifically, we conducted two experiments: (1) a comparison of inference time with several baseline methods in Table 5 and (2) an analysis of the inference time for each component of our CDAM in Table 6. All measurements were performed on an NVIDIA A100 GPU. Note that, due to the shared GPU server environment, the reported inference costs may be higher than those observed on a dedicated local GPU setup.

As shown in Table 5, our proposed CDAM introduces an increase in inference time, approximately 0.026 seconds per image for VOC21, 0.032 seconds per image for Context60, and 0.034 seconds per image for COCO-Obj compared to baseline methods. While CDAM requires additional computational cost, it remains feasible for real-time applications. For example, the inference time of our method (e.g., 0.051 seconds for COCO-Obj with MaskCLIP (Zhou et al., 2022) + CDAM) is significantly

lower than other methods like CaR (Sun et al., 2024) (12.270 seconds for COCO-Obj) and CLIP-DIY (Wysoczańska et al., 2024) (0.559 seconds for COCO-Obj). These results demonstrate that our approach achieves a balance between computational efficiency and enhanced functionality.

Table 5: **Inference time comparison (in seconds per image) of baseline methods with our CDAM.** Despite introducing minimal computational overhead, CDAM remains feasible for real-time applications, especially when compared to computationally intensive methods like CaR and CLIP-DIY.

| Methods | VOC21 | Context60 | COCO-Obj |
|---|---|---|---|
| CaR (Sun et al., 2024) | 3.497 | 9.340 | 12.270 |
| CLIP-DIY (Wysoczańska et al., 2024) | 0.520 | - | 0.559 |
| MaskCLIP (Zhou et al., 2022) | 0.017 | 0.017 | 0.017 |
| MaskCLIP+**CDAM** | 0.043 (+0.026) | 0.049 (+0.032) | 0.051 (+0.034) |
| SCLIP (Wang et al., 2023) | 0.018 | 0.018 | 0.018 |
| SCLIP+**CDAM** | 0.044 (+0.026) | 0.050 (+0.032) | 0.052 (+0.034) |
| ClearCLIP (Lan et al., 2024) | 0.017 | 0.018 | 0.018 |
| ClearCLIP+**CDAM** | 0.044 (+0.027) | 0.050 (+0.032) | 0.051 (+0.033) |
| GEM (Bousselham et al., 2024) | 0.026 | 0.026 | 0.026 sec |
| GEM+**CDAM** | 0.052 (+0.026) | 0.059 (+0.033) | 0.060 (+0.034) |

As shown in Table 6, the majority of the additional computational cost in CDAM arises from its multi-scale image patches ($\text{Attn}_{\text{MS}}$) and augmented text prompts (ATP). The computational overhead increases with the number of scales ($m$) for $\text{Attn}_{\text{MS}}$ and the number of classes for ATP, primarily due to the increased computational burden of calculating the Jensen-Shannon (JS) divergence. Nevertheless, the overall inference time remains within a feasible range, ensuring that CDAM is still practical for real-time applications.

## C  VISUALIZATION OF CLASS DISTRIBUTION-INDUCED ATTENTION MAP

Fig. 5 visualizes examples of class distribution-induced attention maps, highlighting how the attention weight is emphasized on the class-relevant patch for each source patch.

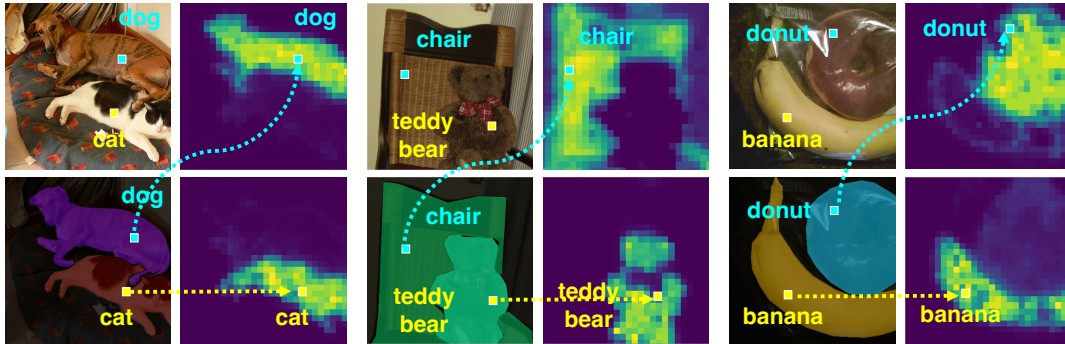

Figure 5: We visualized examples of class distribution-induced attention maps (CDAM) from several source patches. This demonstrates that our proposed CDAM can localize regions of the target class within the attention map.

## D  MODEL DETAILS OF COMPARISON METHODS

For a fair comparison, we provide detailed architectural information and an additional training dataset for all methods, as detailed in Table 7. Importantly, our proposed CDAM can be seamlessly integrated with existing CLIP-based training-free methods, utilizing their CLIP encoder without the need for retraining.

Table 6: **Inference time (in seconds per image) for each component of CDAM.**The baseline model is set as MaskCLIP, as the additional overhead introduced by CDAM is consistent across other baseline methods.

| Baseline | $\text{Attn}_{\text{CDAM}}$ | $\text{Attn}_{\text{MS}}$ | ATP | $\text{Thr}_{\text{ent-bg}}$ | VOC21 | Context60 | COCO-Obj |
|:---:|:---:|:---:|:---:|:---:|:---:|:---:|:---:|
| ✓ | | | | | 0.017 | 0.017 | 0.017 |
| ✓ | ✓ | | | | 0.020 | 0.022 | 0.024 |
| ✓ | ✓ | ✓ | | | 0.032 | 0.038 | 0.040 |
| ✓ | ✓ | ✓ | ✓ | | 0.043 | 0.049 | 0.051 |
| ✓ | ✓ | ✓ | ✓ | ✓ | 0.043 | 0.049 | 0.051 |

Table 7: **Detailed information on comparison methods for open-vocabulary semantic segmentation.** We provide detailed information regarding (1) visual encoder architecture, (2) pre-trained model, and (3) additional training datasets.

| Method | Encoder | Model | Additional training dataset |
|:---|:---|:---|:---|
| *weakly-supervised methods with additional training dataset* | | | |
| GroupViT (Xu et al., 2022a) | ViT/S-16 | Scratch | CC15M+RedCaps12M |
| CLIPpy (Ranasinghe et al., 2023) | ViT/B-16 | CLIP | HQITP-134M |
| ViewCo (Ren et al., 2023) | ViT/S-16 | Scratch | CC12M+YFCC15M |
| SegCLIP (Luo et al., 2023) | ViT/B-16 | CLIP | CC3M+COCO400K |
| OVsegmentor (Xu et al., 2023a) | ViT/B-16 | DINO | CC4M |
| TCL (Cha et al., 2023) | ViT/B-16 | CLIP | CC15M |
| PACL (Mukhoti et al., 2023) | ViT/B-16 | CLIP | CC3M+CC12M+YFCC15M |
| *visual prototype generation methods for each concept* | | | |
| OVDiff (Karazija et al., 2023) | ViT/B-16 | CLIP+SD+DINO | ✗ |
| FreeDA (Barsellotti et al., 2024) | ViT/L-14 | CLIP+SD+DINO | ✗ |
| *CLIP-based training-free methods* | | | |
| CLIPSurgery (Li et al., 2023) | ViT/B-16 | CLIP | ✗ |
| CLIP-DIY (Wysoczańska et al., 2024) | ViT/B-16 | CLIP+DINO | ✗ |
| CaR[†] (Sun et al., 2024) | ViT/L-14 | CLIP | ✗ |
| MaskCLIP[†] (Zhou et al., 2022) | ViT/B-16 | CLIP | ✗ |
| SCLIP[†] (Wang et al., 2023) | ViT/B-16 | CLIP | ✗ |
| ClearCLIP[†] (Lan et al., 2024) | ViT/B-16 | CLIP | ✗ |
| GEM[†] (Bousselham et al., 2024) | ViT/B-16 | CLIP | ✗ |

# E  ADDITIONAL EXPERIMENTS

**Effectiveness of JS divergence.** To assess the effectiveness of JS divergence relative to other metrics like KL divergence and Wasserstein distance (WS), we conducted an ablation study using several baseline methods, as presented in Table 8. The results demonstrate that JS divergence performs comparably to KL divergence, consistently achieving equal or slightly better performance.

**Discussion on the rationale for using JS divergence.** It is reasonable for measuring the similarity of class distributions to have the properties of symmetricity, meaning it should remain the same regardless of input order, and permutation invariance, meaning it should remain the same regardless of class order. The former is satisfied by the Wasserstein distance (WS) and JS divergence, while the latter is satisfied by the KL and JS divergences. Note that it is not straightforward to properly define metrics among classes for the problem of semantic segmentation, making the WS distance unsuitable for our CDAM. Moreover, divergence-based metrics such as KL and JS divergences are known to be more sensitive to small changes over WS distance (Ozair et al., 2019), which may be advantageous for our CDAM. Thus, JS divergence appears to have favorable properties for measuring the distance between class distributions, which is consistent with our experimental validation in Table 8.

**Analysis of CDAM.** As shown in Table 9, we verified the effectiveness of our **$\text{Attn}_{\text{MS}}$** on VOC21 (Everingham et al., 2010), Context60 (Mottaghi et al., 2014) and COCO-Obj (Lin et al., 2014) datasets.

Table 8: **Ablation study of similarity metrics for measuring the distance of class distributions over patches. WS refers to the Wasserstein distance. The evaluation is based on mIoU (%).**

| Methods | Metric | VOC21 | Context60 | COCO-Obj | Avg. |
|---|---|---|---|---|---|
| MaskCLIP (Zhou et al., 2022)+**CDAM** | KL div. | 55.7 | 30.4 | 34.3 | 40.1 |
| | JS div. | 55.9 | 30.5 | 34.3 | **40.2** |
| | WS | 53.0 | 26.7 | 28.5 | 36.1 |
| SCLIP (Wang et al., 2023)+**CDAM** | KL div. | 58.8 | 30.4 | 34.6 | **41.3** |
| | JS div. | 59.0 | 30.4 | 34.5 | **41.3** |
| | WS | 57.2 | 29.0 | 31.4 | 39.2 |
| ClearCLIP (Lan et al., 2024)+**CDAM** | KL div. | 57.4 | 29.5 | 34.3 | 40.4 |
| | JS div. | 57.6 | 29.8 | 34.5 | **40.6** |
| | WS | 56.9 | 28.6 | 33.4 | 39.6 |
| GEM (Bousselham et al., 2024)+**CDAM** | KL div. | 58.9 | 30.5 | 35.1 | **41.5** |
| | JS div. | 58.7 | 30.6 | 35.2 | **41.5** |
| | WS | 58.4 | 29.5 | 34.0 | 40.6 |

Low uncertainty of segmentation for both foreground and background regions corresponds to low and high entropy of the class distribution, respectively. Compared to other types of attention maps, our proposed $\text{Attn}_{MS}$ effectively reduce the uncertainty by widening the gap between $H_{back}$ and $H_{fore}$. This significantly affects the segmentation performance and demonstrates the enhanced localization ability of our $\text{Attn}_{MS}$.

Table 9: **Uncertainty in foreground-background regions.** By varying the types of attention maps in the last layer, we compute the mean entropy of the class distribution in the foreground region ($H_{fore}$), and background region ($H_{back}$), respectively. $\text{Diff}_H$ refers to the absolute difference between $H_{back}$ and $H_{fore}$ ($H_{back} - H_{fore}$). Reducing uncertainty ($\text{Diff}_H \uparrow$) of our proposed CDAM with multi-scale image patches leads to significantly improved segmentation performance.

| | VOC21 | | Context60 | | COCO-Obj | |
|---|---|---|---|---|---|---|
| Attn. Map | $\text{Diff}_H \uparrow$ | mIoU $\uparrow$ | $\text{Diff}_H \uparrow$ | mIoU $\uparrow$ | $\text{Diff}_H \uparrow$ | mIoU $\uparrow$ |
| Random | -0.11 | 8.5 | -0.10 | 4.7 | -0.01 | 4.82 |
| Identity | 0.40 | 33.1 | 0.00 | 23.3 | 0.35 | 24.8 |
| **$\text{Attn}_{MS}$ (ours)** | **1.02** | **54.7** | **0.25** | **27.5** | **1.06** | **34.2** |

**Reproduction details of existing methods.** For a fair comparison among training-free CLIP based methods, we reproduced the prior works, MaskCLIP (Zhou et al., 2022), SCLIP (Wang et al., 2023), CaR (Sun et al., 2024), GEM (Bousselham et al., 2024) and ClearCLIP (Lan et al., 2024), by following the unified evaluation protocol and eliminating the renaming tricks. We remove the expanded the target class names (e.g., "person in shirt", "human" for "person" class ). Then, we selected the results demonstrating the highest overall performance from among those obtained using various background threshold values (0.1, 0.2, 0.3, 0.4, 0.5). An example of the reproduced results is provided in Table 10. Based on the results in Table 10, we set the $\text{Thr}_{default}$ value of SCLIP (Wang et al., 2023) to 0.2 for all benchmark datasets to avoid dataset-specific parameters. For all benchmark datasets, inference was conducted with a sliding window size of 448 and a stride of 224.

**Analysis of entropy-based background threholsding.** Before implementing the entropy-based background thresholding method, we aim to demonstrate its effectiveness. We achieve this by comparing various thresholding strategies with respect to the optimal threshold value that maximizes the mIoU score for a given dataset. To achieve this, we first employ the Pocket algorithm to identify the threshold that maximizes the mIoU value for each image. We then calculate the Pearson correlation coefficient between this optimal threshold and the results obtained using different thresholding techniques. Since Pearson correlation measures the strength of the linear relationship between two variables, we can evaluate which method produces a threshold value most closely aligned with the optimal threshold. The results are presented in Table 11. As evident from the table, our proposed method consistently demonstrates high correlation value with the optimal threshold across most

Table 10: **Reproduced results of SCLIP (Wang et al., 2023) with varying background threshold values.** We evaluated the performance of SCLIP (Wang et al., 2023) with background threshold values ranging from 0.1 to 0.5. The best overall performance was achieved with $\text{Thr}_{\text{default}}$ set to 0.2, and we therefore selected the results obtained at this threshold for the unified evaluatino protocol (Cha et al., 2023). The evaluation is based on mIoU (%).

| $\text{Thr}_{\text{default}}$ | VOC21 | Context60 | COCO-Obj | Avg. |
|---|---|---|---|---|
| 0.1 | 37.8 | **30.5** | 29.9 | 32.7 |
| 0.2 | 50.5 | 25.8 | **31.3** | **35.9** |
| 0.3 | **54.4** | 20.3 | 28.1 | 34.3 |
| 0.4 | 51.8 | 15.7 | 23.4 | 30.3 |
| 0.5 | 46.6 | 11.7 | 18.4 | 25.6 |

datasets. This indicates that there are strong agreement between the optimal threshold and the thresholds generated by our method.

Conversely, other thresholding techniques exhibit either weak or inconsistent correlation with the optimal threshold. Moreover, some techniques might show high correlation on specific datasets, they show downgraded performance in different datasets. In contrast, our method delivers consistently high correlation, highlighting its suitability for this task. This superior performance likely stems from the inherent differences between the target tasks addressed by various methods. Traditional thresholding techniques are primarily designed for grayscale images, which often contain well-defined foreground objects and have higher dimensionality (e.g., 512x512 pixels). Our task, however, presents a greater challenge: thresholding the maximum probability value in images with significantly smaller size (e.g., 28x28 pixels) and uncertain object boundaries. Due to these distinct characteristics, traditional grayscale image thresholding methods are not well-suited for determining the optimal threshold in our specific scenario.

Table 11: **Correlation between optimal threshold and thresholding techniques.** At first, we searched the optimal threshold that maximizes the mIoU score using ground truth. Subsequently, we compare the Pearson correlation coefficient between this optimal threshold and the results obtained using different thresholding strategies. As a result, our proposed method achieves the highest average correlation with the optimal threshold over three benchmark datasets.

| Method | VOC21 | Context60 | COCO-Obj | Avg. |
|---|---|---|---|---|
| Otsu (Otsu et al., 1975) | 0.397 | 0.203 | 0.189 | 0.263 |
| Lloyd (Rosenfeld & De La Torre, 1983) | 0.393 | 0.206 | 0.189 | 0.262 |
| Kittler (Sezan, 1990) | 0.235 | 0.135 | 0.025 | 0.129 |
| Li (Li & Lee, 1993) | 0.280 | 0.207 | 0.055 | 0.181 |
| Kapur (Kapur et al., 1985) | 0.047 | -0.026 | 0.117 | 0.046 |
| Pal (Pal, 1996) | 0.051 | -0.023 | 0.119 | 0.049 |
| Brink (Brink & Pendock, 1996) | 0.147 | 0.058 | -0.129 | 0.025 |
| Huang (Huang & Wang, 1995) | 0.230 | **0.239** | 0.049 | 0.172 |
| **$\text{Thr}_{\text{ent-bg}}$ (Ours)** | **0.477** | 0.175 | **0.321** | **0.325** |

**Tuning of background thresholding value $\text{Thr}_{\text{default}}$ for each dataset.** For each benchmark dataset, we fine-tuned the background thresholding value, $\text{Thr}_{\text{default}}$, and disabled entropy-based background thresholding in our CDAM. As shown in Table 12 , tuning the background thresholding value substantially improves performance compared to the fixed $\text{Thr}_{\text{default}}$ used in the main paper. While existing training-free methods achieve high performance through tuning $\text{Thr}_{\text{default}}$, our proposed CDAM further enhances performance improvements. Notably, our CDAM with MaskCLIP (Zhou et al., 2022) surpasses all existing methods when tuning $\text{Thr}_{\text{default}}$

**Ablation studies for hyperparameter of CDAM.** For the temperature $\tau$, the modulation of entropy $\alpha$ and the set of scaling factor $M$, we conducted an ablation study by varying their values. The baseline model is MaskCLIP (Zhou et al., 2022), and we applied our proposed CDAM with various

Table 12: **Tuning background thresholding value Thr$_{default}$ for CLIP-based training-free methods with CDAM.** We measured the performance of existing training-free methods by varying Thr$_{default}$ from 0.1 to 0.6. Then, we report the highest performance of all methods on three benchmark datasets. We marked [†] for the reproduced methods. Performance improvements by CDAM are indicated in parentheses. The evaluation is based on mIoU (%).

| Methods | VOC21 | Context60 | COCO-Obj | Avg. |
|---|---|---|---|---|
| MaskCLIP[†] (Zhou et al., 2022) | 42.9 | 23.3 | 24.8 | 30.3 |
| MaskCLIP+**CDAM** | 55.0 (+12.1) | **33.6** (+10.3) | 34.4 (+9.6) | 41.0 (+8.3) |
| SCLIP[†] (Wang et al., 2023) | 54.4 | 30.5 | 31.3 | 38.7 |
| SCLIP+**CDAM** | 57.5 (+3.1) | 33.5 (+3.0) | 34.9 (+3.6) | 42.0 (+3.3) |
| ClearCLIP[†] (Lan et al., 2024) | 51.8 | 32.3 | 33.1 | 39.1 |
| ClearCLIP+**CDAM** | 56.3 (+4.5) | 33.0 (+0.7) | 34.6 (+1.5) | 41.3 (+2.2) |
| GEM[†] (Bousselham et al., 2024) | 53.8 | 32.4 | 34.1 | 40.1 |
| GEM+**CDAM** | **57.3** (+3.5) | **33.6** (+1.2) | **35.6** (+1.5) | **42.2** (+2.1) |

hyperparameters.Based on the results shown in Table 13, 14, 15, we set $\tau$, $\alpha$ and $M$ to 0.1, 2.5 and (0.25, 0.37, 0.5, 0.63, 0.75, 0.87, 1.0), respectively.

Table 13: **Ablation study on temperature $\tau$ for generating the class distribution-induced attention map.**

| $\tau$ | VOC21 | Context60 | COCO-Obj | Avg. |
|---|---|---|---|---|
| 0.05 | 53.3 | 29.6 | 33.0 | 38.6 |
| 0.10 (**Ours**) | **55.9** | **30.5** | **34.3** | **40.2** |
| 0.15 | 55.6 | 29.7 | 33.6 | 39.6 |
| 0.20 | 52.5 | 28.1 | 31.1 | 37.2 |

Table 14: **Ablation study on modulation parameter $\alpha$ for entropy-based background thresholding.**

| $\alpha$ | VOC21 | Context60 | COCO-Obj | Avg. |
|---|---|---|---|---|
| 1.0 | 44.1 | **33.5** | 28.6 | 35.4 |
| 1.5 | 50.5 | 33.3 | 31.8 | 38.5 |
| 2.5 (**Ours**) | **55.9** | 30.5 | **34.3** | **40.2** |
| 3.5 | 48.3 | 25.2 | 34.0 | 35.8 |

Table 15: **Ablation study on set of scaling factor $M$ for the CDAM with multi-scale image patches.**

| Set of scaling factor $M$ | VOC21 | Context60 | COCO-Obj | Avg. |
|---|---|---|---|---|
| (0.25,0.37,0.50,0.63,0.75,0.87,1.00) (**Ours**) | **55.9** | **30.5** | 34.3 | **40.2** |
| (0.50,0.63,0.75,0.87,1.00) | 55.8 | 30.4 | **34.4** | **40.2** |
| (0.75,0.87,1.00) | 55.5 | 30.2 | 34.2 | 40.0 |
| (1.00) | 53.1 | 27.7 | 32.1 | 37.6 |
| (0.25,0.50,0.75,1.00) | 55.7 | 30.4 | 34.1 | 40.1 |

**Ablation studies for different patch size.** To explore the impact of different patch sizes on our proposed CDAM, we conducted benchmark experiments using CLIP ViT/B-32, which has a patch size of 32. Note that the results presented in the main paper used CLIP ViT/B-16, which uses a patch

size of 16. As shown in the Table 16, these results demonstrate that our proposed CDAM effectively enhances the performance of CLIP-based training-free methods, even when using CLIP models with different and larger patch sizes. It also shows that a larger patch size (32) typically results in decreased performance compared to a smaller patch size (16) in Table 1 of the main paper. This suggests that small patch size of 16 is advantageous over large patch size of 32 for segmentation due to its ability to provide better spatial resolution. In this experiment, the modulation of entropy, $\alpha$, is set to 2.0.

Table 16: **Ablation study with CLIP ViT/B-32 for exploring different patch sizes.** We evaluated the open-vocabulary semantic segmentation performance of existing training-free methods. We marked $^\dagger$ for the reproduced methods. Performance improvements achieved by CDAM are indicated in parentheses. The evaluation metric used is mIoU (%).

| Methods | VOC21 | Context60 | COCO-Obj | Avg. |
|---|---|---|---|---|
| MaskCLIP$^\dagger$ (Zhou et al., 2022) | 29.5 | 8.1 | 11.5 | 16.4 |
| MaskCLIP+**CDAM** | 50.1 (+20.6) | 27.6 (+19.5) | 27.8 (+16.3) | 35.2 (+18.8) |
| SCLIP$^\dagger$ (Wang et al., 2023) | 38.0 | 24.1 | 25.1 | 29.1 |
| SCLIP+**CDAM** | 51.6 (+13.6) | 25.7 (+1.6) | 27.6 (+2.5) | 35.0 (+5.9) |
| ClearCLIP$^\dagger$ (Lan et al., 2024) | 47.6 | 23.3 | 27.3 | 32.7 |
| ClearCLIP+**CDAM** | 51.4 (+3.8) | 27.5 (+4.2) | 28.4 (+1.1) | 35.8 (+3.1) |
| GEM$^\dagger$ (Bousselham et al., 2024) | 52.1 | 28.1 | 33.8 | 38.0 |
| GEM+**CDAM** | **55.9** (+3.8) | **32.4** (+4.3) | **34.4** (+0.6) | **40.9** (+2.9) |

**Supporting our motivation with several baseline methods.** As mentioned in Section 3.2.1, Table 17 provides statistical evidence to support our claim "CLIP-based prior works yield patch-wise noisy class predictions while having highly correlated class distributions for each object".

The experimental setup is as follows: For a given image, we randomly selected $P_{target}$ within the target class region and also randomly selected $P_{in}$ and $P_{target}$ from the the target class region and the rest of the region, respectively. We then measured (1) the probability that the class prediction for $P_{target}$ is correct and (2) the probability that the distribution similarity between $P_{target}$ and $P_{in}$ is higher than the distribution similarity between $P_{target}$ and $P_{out}$.

Table 17: **Accuracy comparison of class predictions and similarity of class distributions with several CLIP-based training-free methods across datasets.** Similarity of class distribution is measured using JS divergence.

| Baseline | | VOC21 | Context60 | COCO-Obj | Avg. |
|---|---|---|---|---|---|
| MaskCLIP (Zhou et al., 2022) | Class Prediction | $56.1 \pm 1.17$ | $38.4 \pm 0.23$ | $27.4 \pm 0.46$ | 43.0 |
| | Sim of Class Dist | $70.9 \pm 0.44$ | $73.1 \pm 0.34$ | $69.8 \pm 0.42$ | 71.0 |
| SCLIP (Wang et al., 2023) | Class Prediction | $67.0 \pm 0.49$ | $41.8 \pm 0.31$ | $33.6 \pm 0.23$ | 47.4 |
| | Sim of Class Dist | $78.9 \pm 0.26$ | $72.0 \pm 0.30$ | $75.4 \pm 0.55$ | 75.5 |
| ClearCLIP (Lan et al., 2024) | Class Prediction | $70.3 \pm 0.51$ | $42.7 \pm 0.23$ | $36.4 \pm 0.19$ | 49.9 |
| | Sim of Class Dist | $76.0 \pm 0.58$ | $70.5 \pm 0.58$ | $71.7 \pm 0.48$ | 72.3 |
| GEM (Bousselham et al., 2024) | Class Prediction | $70.8 \pm 1.12$ | $42.4 \pm 0.38$ | $37.5 \pm 0.45$ | 50.0 |
| | Sim of Class Dist | $79.4 \pm 0.89$ | $71.2 \pm 0.34$ | $74.2 \pm 0.20$ | 74.8 |

**Analysis of performance discrepancy between SCLIP and ClearCLIP with CDAM.** In Table 1 and 2 of the main paper, the performance of ClearCLIP (Lan et al., 2024) surpasses that of SCLIP (Wang et al., 2023). However, when combined with CDAM, SCLIP outperforms ClearCLIP. This discrepancy arises from ClearCLIP's design choice to remove residual connections to suppress noisy class predictions, which becomes less effective when used alongside CDAM. The absence of residual connections in ClearCLIP, combined with the lack of additional training, likely hinders its ability to fully utilize CLIP's implicit knowledge of class distributions. Consequently, the synergy between CDAM and ClearCLIP is reduced, resulting in smaller performance gains relative to SCLIP.

**Applying our CDAM on mask-annotation based supervision methods.** We conducted additional experiments to validate the effectiveness of our CDAM by incorporating it into existing methods that

use mask annotations for specific classes in a supervised manner. Specifically, we applied CDAM to mask-annotation based methods and report the results in Table 18. As demonstrated, our CDAM did not degrade the performance of seen classes optimized by training-based methods with mask annotation such as ZegCLIP (Zhou et al., 2023) and OTSeg (Kim et al., 2024). Instead, it consistently improved performance for unseen classes.

Table 18: **Results of open-vocabulary semantic segmentation methods supervised using mask annotations with our CDAM on VOC 2012.** hIoU refers to harmonic mean IoU. Performance improvements by CDAM are indicated in parentheses.

| Method | mIoU (Seen) | mIoU (Unseen) | hIoU |
|---|---|---|---|
| ZegCLIP (Zhou et al., 2023) | 91.8 | 77.9 | 84.3 |
| ZegCLIP+**CDAM** | 91.8 (+0.0) | 78.4 (+0.5) | 84.6 (+0.3) |
| OTSeg (Kim et al., 2024) | 93.3 | 81.8 | 87.2 |
| OTSeg+**CDAM** | **93.4** (+0.1) | **82.2** (+0.4) | **87.4** (+0.2) |

## F  QUALITATIVE RESULTS

We visualize additional qualitative segmentation results in Fig. 6. It demonstrates that accurate localization ability of our CDAM for semantic segmentation. Furthermore, in Fig. 7, we present more examples of class distribution-induced attention maps. It indicates that without the necessity of training and annotation, we can construct the well-localized attention map based on the similarity of class distribution between patches within an image.

## G  LIMITATIONS

Leveraging the knowledge of pre-trained vision-language models, we construct a class distribution-induced attention map based on class distribution of target classes. However, the quality of the localized attention map is dependent on the diversity and number of target classes used. This dependence can limit the representation of CLIP's implicit knowledge within the class distribution. Specifically, CLIP's limitations in class diversity and image-text representation hinder segmentation performance, particularly for rare classes. CLIP-based training-free methods inherit these constraints, as they lack additional training. While CDAM improves the performance of these approaches, it cannot overcome CLIP's fundamental limitations. Addressing such challenges may require training-based strategies like domain-specific prompt tuning or advanced multi-modal models with better class diversity and image-text alignment. However, fully resolving out-of-distribution (OOD) issues or achieving complete class coverage remains challenging without extensive training, which is beyond the scope of this work.

Entropy-based background thresholding, being empirically designed, often struggles in handling complex and challenging test images. The primary difficulty lies in its inability to determine patch-specific threshold values effectively in complex scenes. In simpler scenarios, such as when a single target class object is prominently located at the center of the image, a single threshold value may suffice. However, as scene complexity increases, local thresholding becomes necessary, which introduces additional challenges. To address these limitations and improve the robustness of background detection, data-driven methods like FOUND (Siméoni et al., 2023) offer a potential solution. FOUND proposed an unsupervised framework to effectively extract background regions. However, its dependence on saliency-based foreground detection, without incorporating knowledge of target classes, limits its adaptability for open-vocabulary semantic segmentation. While data-driven approaches show promise, achieving universally robust and adaptable background thresholding remains a significant challenge.

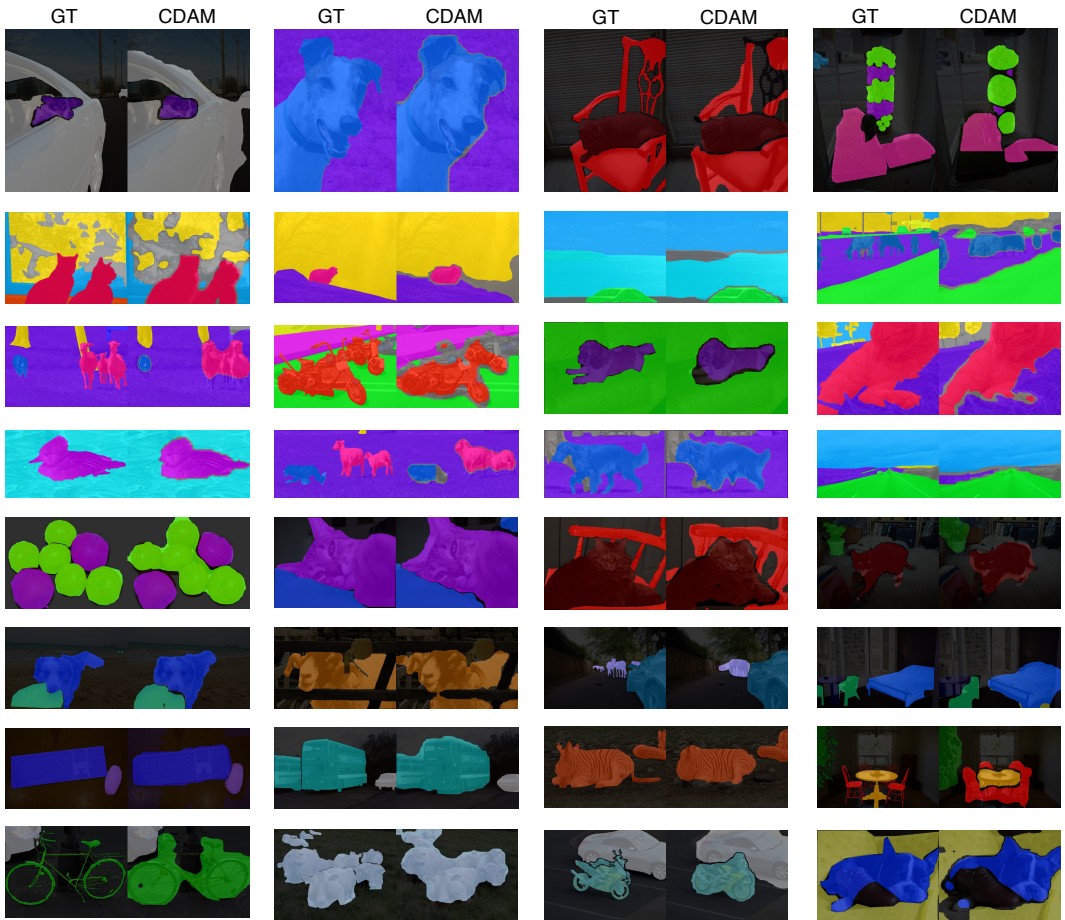

Figure 6: **Qualitative results of open-vocabulary semantic segmentation using our CDAM with MaskCLIP.**

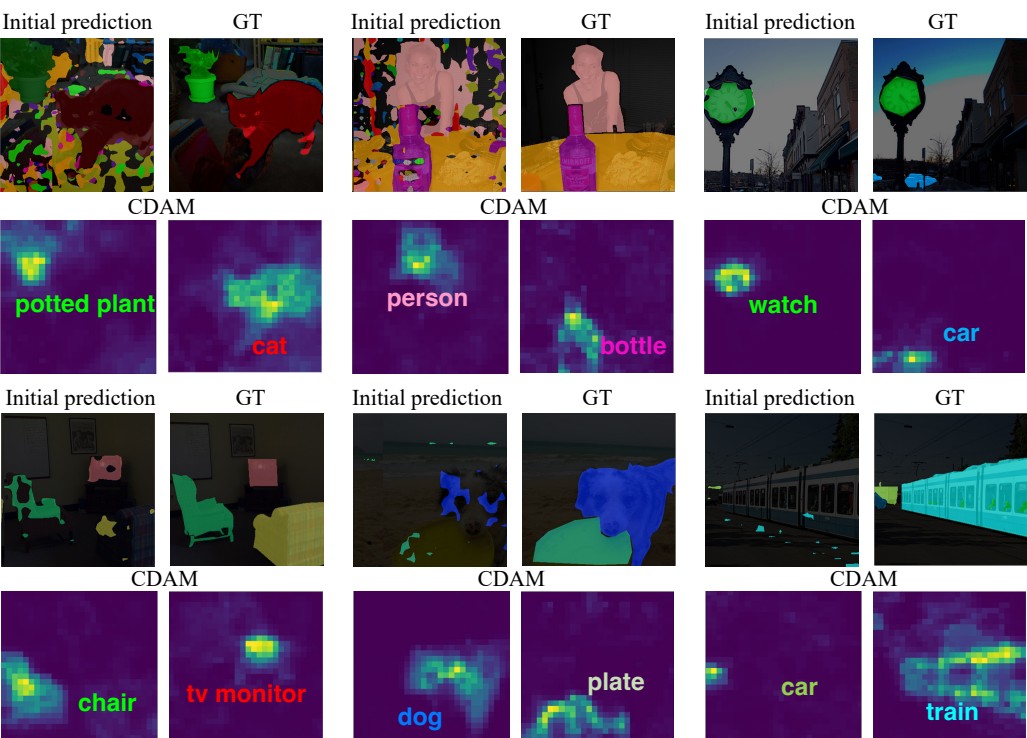

Figure 7: **Additional examples of class distribution-induced attention maps (CDAM) from the initial prediction of MaskCLIP.**

