# OpenReview forum: "Class Distribution-induced Attention Map for Open-vocabulary Semantic Segmentations"
_ICLR.cc/2025/Conference — ICLR 2025 Poster_

### Official Review · Reviewer_sGKo · 2024-10-21

**Soundness:** 3
**Presentation:** 2
**Contribution:** 2
**Rating:** 6
**Confidence:** 4

**Summary:**

This paper introduces CDAM for open-vocabulary semantic segmentation that improves object localization by utilizing class distribution similarities within patches. It employs Jensen-Shannon divergence to create an attention map that boosts CLIP's segmentation accuracy without extra training. CDAM also uses multi-scale patches and entropy-based thresholding for enhanced performance, outperforming other methods on segmentation benchmarks.

**Strengths:**

1. This paper improves the localization of objects within images by leveraging class distribution similarities.

2. The entropy-based background thresholding adapts dynamically to different images, which helps in accurately separating the foreground from the background in segmentation tasks.

**Weaknesses:**

1. There is little explanation why the Jensen-Shannon divergence is suitable for semantic segmentation. The rationality of this method should be explained clearly.

2. There are some confusing aspects in the use of symbols in this paper, such as the final similarity map S in line 251 and the class distribution S in line 266; The image representation is unclear, for example, the meaning of S_p1 in Figure 1 has not been mentioned yet in the paper.

3. The JS divergence is used in the paper to obtain Attn_CDAM.  How is the effect of using KL divergence and other measurement methods.

4. What is the inference time after adding CDAM to other methods.

5. The word 'food' appears twice in the Supercategory in Tab 4. An additional ‘()’ appeared in line 330

**Questions:**

1. How to explain the rationality of using two hyper-parameters α and Thr_default to control the background thresholding in Formula 4?

**Details Of Ethics Concerns:**

There is no ethics concerns.

---

> ### Author Response · Authors · 2024-11-22
>
> We appreciate your insightful feedback and comments.
>
> > **[W1] There is little explanation why the Jensen-Shannon divergence is suitable for semantic segmentation. The rationality of this method should be explained clearly.**
>
> The Jensen-Shannon (JS) divergence is particularly suitable to construct our CDAM due to its symmetry and bounded range, which enhance its robustness compared to other metrics, such as KL divergence and Wasserstein distance, in generating attention maps. Specifically, KL divergence is asymmetric and has an unbounded maximum value, while Wasserstein distance, although symmetric, also lacks a bounded range. Empirically, we have demonstrated that the JS divergence is more effective for open-vocabulary semantic segmentation compared to these alternatives (see our response to [W3] below with a table for further details).
>
> > **[W3] The JS divergence is used in the paper to obtain Attn_CDAM. How is the effect of using KL divergence and other measurement methods.**
>
> We have conducted an ablation study using several baseline methods with diverse metrics such as Kullback–Leibler (KL) divergence and Wasserstein distance (WS). Table 3 demonstrates that JS divergence consistently achieves the best averaged performance across three benchmarks. We chose JS divergence due to its symmetricity and now Table 3 supports our choice of JS divergence sufficiently. We have updated these results in revised paper (Line 465-468 and Line 890-893).
>
> **Table 1:** Ablation study of similarity metrics for measuring the distance of class distributions over patches. WS refers to the Wasserstein distance. The evaluation is based on mIoU(\%).
>
> |Method|Metric|VOC21|Context60|COCO-Obj|Avg.|
> |-|-|-|-|-|-|
> |MaskCLIP+CDAM|KL|55.7|30.4|34.3|40.1|
> ||JS|55.9|30.5|34.3|**40.2**|
> ||WS|53.0|26.7|28.5|36.1|
> |SCLIP+CDAM|KL|58.8|30.4|34.6|**41.3**|
> ||JS|59.0|30.4|34.5|**41.3**|
> ||WS|57.2|29.0|31.4|39.2|
> |ClearCLIP+CDAM|KL|57.4|29.5|34.3|40.4|
> ||JS|57.6|29.8|34.5|**40.6**|
> ||WS|56.9|28.6|33.4|39.6|
> |GEM+CDAM|KL|58.9|30.5|35.1|**41.5**|
> ||JS|58.7|30.6|35.2|**41.5**|
> ||WS|58.4|29.5|34.0|40.6|
>
> > **[W2] There are some confusing aspects in the use of symbols in this paper, such as the final similarity map S in line 251 and the class distribution S in line 266; The image representation is unclear, for example, the meaning of S_p1 in Figure 1 has not been mentioned yet in the paper.**
>
> In the revision, we have unified the terminology throughout the paper, consistently referring to it as "the similarity map $S$." Additionally, $S_{P1}$ represents the class distribution at the position of patch 1 ($P1$) within the similarity map $S$. We have clarified this in the figure caption to ensure better understanding.

---

> ### Author Response · Authors · 2024-11-22
>
> > **[W4] What is the inference time after adding CDAM to other methods.**
>
> We have conducted two experiments (1 NVIDIA A100 GPU used): (1) a comparison of inference time with several baselines (Table 2) and (2) an analysis of the inference time for each component of our CDAM (Table 3). Considering how much CDAM can improve performance (to the level of CaR!), the additional computation time can be justified and one can still obtain the result for one image within at most 60 msec (0.06 sec), which may still suitable for many real-time applications. Note that Table 5 shows that the majority of the computational cost in CDAM arises from its multi-scale image patches ($\text{Attn}\_\text{MS}$) and augmented text prompts (ATP). The computational overhead increases with the number of scales ($m$) for $\text{Attn}_\text{MS}$ and the number of classes for ATP, primarily due to the increased computational burden of computing Jensen-Shannon (JS) divergence. These results have been included in the revised paper (Line 460-464 and Line 782-804).
>
> **Table 2:** Inference time comparison (in seconds per image) of baseline methods with our CDAM.
>
> |Method|VOC21|Context60|COCO-Obj|
> |-|-|-|-|
> |CaR|3.497 sec|9.340 sec|12.270 sec|
> |CLIP-DIY|0.520 sec|-|0.559 sec|
> |MaskCLIP|0.017 sec|0.017 sec|0.017 sec|
> |MaskCLIP+**CDAM**|0.043 sec (+0.026)|0.049 sec (+0.032)|0.051 sec (+0.034)|
> |SCLIP|0.018 sec|0.018 sec|0.018 sec|
> |SCLIP+**CDAM**|0.044 sec (+0.026)|0.050 sec (+0.032)|0.052 sec (+0.034)|
> |ClearCLIP|0.017 sec|0.018 sec|0.018 sec|
> |ClearCLIP+**CDAM**|0.044 sec (+0.027)|0.050 sec (+0.032)|0.051 sec (+0.033)|
> |GEM|0.026 sec|0.026 sec|0.026 sec|
> |GEM+**CDAM**|0.052 sec (+0.026)|0.059 sec (+0.033)|0.060 sec (+0.034)|
>
>
> **Table 3:** Inference time (in seconds per image) for each component of CDAM. The baseline model is set as MaskCLIP, as the additional overhead introduced by CDAM is consistent across other baseline methods. Ba denotes Baseline, At-C denotes $\text{Attn} _\text{CDAM}$, At-M denotes $\text{Attn} _\text{MS}$, ATP refers to the augmented text prompts and Th-e denotes $\text{Thr} _\text{ent-bg}$.
>
> |Ba|At-C|At-M|ATP|Th-e|VOC21|Context60|COCO-Obj
> |-|-|-|-|-|-|-|-|
> |✔| | | | |0.017 sec|0.017 sec|0.017 sec|
> |✔|✔| |||0.020 sec|0.022 sec|0.024 sec|
> |✔|✔|✔|||0.032 sec|0.038 sec|0.040 sec|
> |✔|✔|✔|✔||0.043 sec|0.049 sec|0.051 sec|
> |✔|✔|✔|✔|✔|0.043 sec|0.049 sec|0.051 sec|
>
>
> > **[W5] The word 'food' appears twice in the Supercategory in Tab 4. An additional ‘()’ appeared in line 330**
>
> We have resolved the issues you mentioned.
>
> > **[Q1] How to explain the rationality of using two hyper-parameters α and Thr_default to control the background thresholding in Formula 4?**
>
> As mentioned in the first paragraph of Section 3.3, in an ideal case where the segmentation model performs well, the background class can be differentiated using a default threshold value ($\text{Thr} _\text{default}$) of 0.5. For this reason, $\text{Thr} _\text{defualt}$ was set to 0.5 in FreeDA (Barsellotti et al., CVPR 2024), and we have also fixed this value at 0.5 in our approach. However, to account for the potential inaccuracies of the segmentation model, Formula 4 introduces the use of $\alpha$ / $\text{H}(\text{S}) _\text{center}$ to adaptively control $\text{Thr} _\text{default}$. Specifically, $\alpha$ serves as a scaling constant for the value of $\text{H}(\text{S}) _\text{center}$.

---

> > ### Comment · Reviewer_sGKo · 2024-11-24
> >
> > The authors' responses are greatly appreciated. As most of my concerns are resolved, I would raise my score.

---

> ### Author Response · Authors · 2024-11-27
>
> We are very pleased that most of your concerns have been addressed and sincerely appreciate the increase in the score. The reviewer's insights and suggestions have been invaluable in improving the clarity and quality of our work. Thank you once again for your valuable feedback.

---

### Official Review · Reviewer_6UEY · 2024-11-01

**Soundness:** 3
**Presentation:** 2
**Contribution:** 3
**Rating:** 6
**Confidence:** 4

**Summary:**

This manuscript is dealing with Open-vocabulary semantic segmentation aiming to improve the labeling of  individual pixels with both seen and unseen classes, with a focus on localization and background separation. This work leverages class distribution comparisons between patches of the same object to improve localization. By integrating their approach (CDAM) into CLIP’s final layer, the model’s ability to focus on desired classes is enhanced. CDAM also supports multi-scale image patches and augmented text prompts, improving segmentation accuracy and enabling entropy-based background thresholding. The presented results show some performance improvements on standard benchmarks.

**Strengths:**

The idea of the manuscript is rather simple and straight-forward. Nevertheless, there are some things to be highlighted:
1. By using class distribution similarities the approach aims at enhancing the localization of objects in open-vocabulary segmentation  addressing up to a point the problem of other methods when coping with the patch-wise noise in class predictions. A more reliable attention map is statistically obtained by looking at the Jensen-Shannon divergence between patch pairs belonging to the same class..
2. Although not new, it is good that the proposed approach is able to deal with multi-scale patches and augmented prompts making it a more versatile framework able to more precisely capture class distinctions. Along the same lines (not really new) is the entropy-based background thresholding enhancing the segmentation performance by providing a cleaner separation of relevant classes from the background.
3. Probably the most important contribution is the compatibility with other zero-shot segmentation approaches allowing the approach to be integrated with existing models, potentially compounding improvements without much redundancy. This adaptability offers an approach that complements, rather than competes with, prior work, making it suitable for further extensions.

**Weaknesses:**

There are several issues that need to be clarified.
1. Increased Computational Complexity. There is little discussion regarding the complexity overhear introduced, especially with the Jensen-Shannon divergence calculation and multi-scale patch analysis that might require significant computational resources, which could limit its scalability in real-time or resource-constrained applications.
2. There is an inherent dependency on the CLIP Model inheriting the CLIP’s limitations in terms of class diversity, image-text representation, and the specificity of semantic segmentation. Any inherent biases or limitations in the CLIP model could be amplified or remain unresolved in this framework.
3. The background thresholding approach is rather heuristic (also indicated by the authors) and this brings uncertainty regarding its robustness especially in the presence of highly complex scenes with ambiguous background features. Accurately setting thresholds for diverse and dynamic backgrounds could be challenging and might require extensive tuning for different environments.
4. It is unclear why most of the detailed analysis has been done on top of MaskCLIP and not on some of the newer approaches. I understand that the improvement is larger when compared to MaskCLIP but one would have expected to see the analysis on the more performing approaches. Some of the details are missing or the authors are treated them superficially. For example when doing the analysis of the results in Table 1 they simply indicate that the best performing approach, i.e., CaR requires high computational costs but it is unclear why this is indeed a problem given that CDAM is supposed to be added on top of it.
5. There is a no insight into generalization on rare or fine-grained classes. The approach emphasizes improvement in localization but may not specifically address challenges in recognizing rare or highly similar fine-grained classes, a common difficulty in open-vocabulary segmentation.
6. Although CDAM is designed to work alongside existing zero-shot methods, effectively combining this technique with other methods might be challenging in practice. There is practically no discussion highlighting for example whether this compatibility would likely require additional tuning and could complicate model training, implementation, and maintenance.

**Questions:**

The questions below are summarizing the weaknesses I've highlighted above.
1. How does the complexity introduced by the Jensen-Shannon divergence calculation and multi-scale patch analysis impact the model’s runtime and feasibility in real-time applications?
2. To what extent do CLIP’s limitations in class diversity and image-text representation influence the segmentation results of CDAM?
3, What specific challenges could arise when applying the heuristic entropy-based background thresholding in complex scenes with ambiguous background features? Are there data-driven approaches that could replace the heuristic thresholding method to improve robustness and adaptability?
4.  Why did the authors focus their analysis primarily on MaskCLIP, and what advantages or disadvantages does this bring in evaluating CDAM’s effectiveness? How would CDAM’s performance and computational requirements compare if implemented on more recent segmentation models like CaR or others?
5. How might the CDAM approach be adapted to address challenges in recognizing rare or fine-grained classes that are crucial in open-vocabulary segmentation?

---

> ### Author Response · Authors · 2024-11-22
>
> We appreciate the reviewer's valuable and insightful feedbacks.
>
> > **[Q1] How does the complexity introduced by the Jensen-Shannon divergence calculation and multi-scale patch analysis impact the model’s runtime and feasibility in real-time applications?**
>
> We have conducted two experiments (1 NVIDIA A100 GPU used): (1) a comparison of inference time with several baselines (Table 1) and (2) an analysis of the inference time for each component of our CDAM (Table 2). Considering how much CDAM can improve performance (to the level of CaR!), the additional computation time can be justified and one can still obtain the result for one image within at most 60 msec (0.06 sec), which may still suitable for many real-time applications. Note that Table 2 shows that the majority of the computational cost in CDAM arises from its multi-scale image patches ($\text{Attn}\_\text{MS}$) and augmented text prompts (ATP). The computational overhead increases with the number of scales ($m$) for $\text{Attn}_\text{MS}$ and the number of classes for ATP, primarily due to the increased computational burden of computing Jensen-Shannon (JS) divergence. These results have been included in the revised paper (Line 460-464 and Line 782-804).
>
> **Table 1:** Inference time comparison (in seconds per image) of baseline methods with our CDAM. Despite introducing minimal computational overhead, CDAM remains feasible for real-time applications, especially when compared to computationally intensive methods like CaR and CLIP-DIY.
>
> |Method|VOC21|Context60|COCO-Obj|
> |-|-|-|-|
> |CaR|3.497 sec|9.340 sec|12.270 sec|
> |CLIP-DIY|0.520 sec|-|0.559 sec|
> |MaskCLIP|0.017 sec|0.017 sec|0.017 sec|
> |MaskCLIP+**CDAM**|0.043 sec (+0.026)|0.049 sec (+0.032)|0.051 sec (+0.034)|
> |SCLIP|0.018 sec|0.018 sec|0.018 sec|
> |SCLIP+**CDAM**|0.044 sec (+0.026)|0.050 sec (+0.032)|0.052 sec (+0.034)|
> |ClearCLIP|0.017 sec|0.018 sec|0.018 sec|
> |ClearCLIP+**CDAM**|0.044 sec (+0.027)|0.050 sec (+0.032)|0.051 sec (+0.033)|
> |GEM|0.026 sec|0.026 sec|0.026 sec|
> |GEM+**CDAM**|0.052 sec (+0.026)|0.059 sec (+0.033)|0.060 sec (+0.034)|
>
> **Table 2:** Inference time (in seconds per image) for each component of CDAM. The baseline model is set as MaskCLIP, as the additional overhead introduced by CDAM is consistent across other baseline methods. Ba denotes Baseline, At-C denotes $\text{Attn} _\text{CDAM}$, At-M denotes $\text{Attn} _\text{MS}$, ATP refers to the augmented text prompts and Th-e denotes $\text{Thr} _\text{ent-bg}$.
>
> |Ba|At-C|At-M|ATP|Th-e|VOC21|Context60|COCO-Obj
> |-|-|-|-|-|-|-|-|
> |✔| | | | |0.017 sec|0.017 sec|0.017 sec|
> |✔|✔| |||0.020 sec|0.022 sec|0.024 sec|
> |✔|✔|✔|||0.032 sec|0.038 sec|0.040 sec|
> |✔|✔|✔|✔||0.043 sec|0.049 sec|0.051 sec|
> |✔|✔|✔|✔|✔|0.043 sec|0.049 sec|0.051 sec|

---

> ### Author Response · Authors · 2024-11-22
>
> > **[Q2.1] To what extent do CLIP’s limitations in class diversity and image-text representation influence the segmentation results of CDAM?**
>
> CLIP’s limitations in class diversity and image-text representation hinder segmentation performance in general, especially with rare classes in real-world scenarios. These challenges are common to training-free methods (e.g., SCLIP, ClearCLIP, GEM), as they inherit the CLIP's constraints without additional training. CDAM itself is not a segmentation method, but a method to aid other CLIP-based methods for enhancing the performance of segmentation. Thus, CDAM may be able to fullfil its goal (i.e., improvement over the original method) despite CLIP's limitations that are shared with the original method.
>
> However, CDAM won't be able to overcome CLIP's fundamental limitations. We believe that to overcome these issues, training-based approaches such as domain-specific prompt tuning and utilizing more advanced multi-modal foundation models with enhanced class diversity and stronger image-text alignment could mitigate some of these limitations. However, it’s important to note that fully addressing out-of-distribution (OOD) scenarios or achieving perfect class coverage remains challenging without extensive training. CDAM's role with the training-based methods is beyond the scope of this training-free work.
>
> > **[Q2.2] What specific challenges could arise when applying the heuristic entropy-based background thresholding in complex scenes with ambiguous background features? Are there data-driven approaches that could replace the heuristic thresholding method to improve robustness and adaptability?**
>
> The most significant challenge with complex scenes lies in its inability to determine patch-wise threshold values effectively. For simpler scenes (e.g., where a single target-class object is prominently located in the center of the image), using a single threshold value across the entire image may suffice. However, as scenes become more complex, local thresholding becomes necessary, which introduces significant challenges. While our proposed thresholding method has shown excellent performance over VOC21, Context60 and COCO-Obj, leveraging Multi Otsu algorithm to modify our proposed method can have a good potential for more challenging scenes. However, the immediate challenge for this extension lies in secure related datasets with complex scenes and corresponding complex annotations, which is beyond the scope of this work.
>
> To address these challenges and enhance the robustness of background detection, data-driven approaches like FOUND (Simeoni et al., CVPR 2023), utilized in CLIP-DIY, can be considered. FOUND leverages DINO in an unsupervised framework to effectively extract background regions. However, its reliance on saliency-based foreground detection without knowledge of target classes limits its adaptability in open-vocabulary semantic segmentation. While data-driven methods show promise, achieving robust and universally adaptable background thresholding across all scenarios remains a significant challenge.

---

> ### Author Response · Authors · 2024-11-22
>
> > **[Q3] Why did the authors focus their analysis primarily on MaskCLIP, and what advantages or disadvantages does this bring in evaluating CDAM’s effectiveness? How would CDAM’s performance and computational requirements compare if implemented on more recent segmentation models like CaR or others?**
>
> We chose MaskCLIP initially since it was a well-known method in the community (468 citations). To relieve your concern, we have performed further analysis with more recent baselines, which was reported in Table 3 below. Table 3 provides statistical evidence to support our claim over diverse recent baselines, reporting the results of the following experiment: For a given image, one patch $P_{target}$ was randomly selected and then two patches $P_{in}$ and $P_{out}$ were randomly selected from the target class region and the rest of the region, respectively. Then, we measure (1) the probability of {class prediction in $P_{target}$ is correct} and (2) the probability of {distribution similarity between $P_{target}$ and $P_{in}$ < distribution similarity between $P_{target}$ and $P_{out}$}. These results clearly support that our claim is still valid even for more recent CLIP-based prior works such as SCLIP, ClearCLIP and GEM. These results were reflected in the revision (Line 211-221).
>
> In addition, we have conducted an ablation study to validate the effectiveness of our proposed CDAM components over various baselines including recent works such as SCLIP, ClearCLIP, and GEM. Table 4 shows that the addition of our CDAM components consistently improved open-vocabulary semantic segmentation performance without requiring additional training. These new studies strengthen our claim and demonstrate the robustness and novelty of our approach over diverse prior arts. The corresponding updates have been incorporated into the revised paper (Section 4.3)
>
> Lastly, many recent CLIP-based training-free classification methods rely on task-agnostic local visual tokens and our CDAM can be seamlessly integrated with them for improved performance. However, CaR and CLIP-DIY use task-specific CLS tokens for region classification, so it is not straightforward to incorporate them with our CDAM yet. We have included this clarification in the revised paper (Line 377-405).
>
> **Table 3:** Accuracy comparison of class predictions and similarity of class distributions with several CLIP-based training-free methods across datasets. Similarity of class distribution is measured using JS divergence.
>
> |Baseline||VOC21|Context60|COCO-Obj|Avg.|
> |-|-|-|-|-|-|
> |MasKCLIP|Class Prediction|56.1 $\pm$ 1.17|38.4 $\pm$ 0.23|27.4 $\pm$ 0.46|43.0|
> ||Sim of Class Dist|70.9 $\pm$ 0.44|73.1 $\pm$ 0.34|69.8 $\pm$ 0.42|71.0|
> |SCLIP|Class Prediction|67.0 $\pm$ 0.49|41.8 $\pm$ 0.31|33.6 $\pm$ 0.23|47.4|
> ||Sim of Class Dist|78.9 $\pm$ 0.26|72.0 $\pm$ 0.30|75.4 $\pm$ 0.55|75.5|
> |ClearCLIP|Class Prediction|70.3 $\pm$ 0.51|42.7 $\pm$ 0.23|36.4 $\pm$ 0.19|49.9|
> ||Sim of Class Dist|76.0 $\pm$ 0.58|70.5 $\pm$ 0.58|71.7 $\pm$ 0.48|72.3|
> |GEM|Class Prediction|70.8 $\pm$ 1.12|42.4 $\pm$ 0.38|37.5 $\pm$ 0.45|50.0|
> ||Sim of Class Dist|79.4 $\pm$ 0.89|71.2 $\pm$ 0.34|74.2 $\pm$ 0.20|74.8|
>
>
> **Table 4:**  Ablation study on components of our CDAM with several baseline methods. We measured the performance on VOC21. ATP refers to the augmented text prompts. Ba denotes Baseline, At-C denotes $\text{Attn} _\text{CDAM}$, At-M denotes $\text{Attn} _\text{MS}$, ATP refers to the augmented text prompts and Th-e denotes $\text{Thr} _\text{ent-bg}$. The evaluation is based on mIoU(\%).
>
> |Ba|At-C|At-M|ATP|Th-e|MaskCLIP|SCLIP|ClearCLIP|GEM|
> |-|-|-|-|-|-|-|-|-|
> |✔| | | | |33.1|50.5|50.7|52.1|
> |✔|✔| |||50.1|55.0|52.1|54.7|
> |✔|✔|✔|||53.7|56.9|55.8|56.5|
> |✔|✔|✔|✔||54.7|57.2|56.0|56.9|
> |✔|✔|✔|✔|✔|**55.9**|**59.0**|**57.6**|**58.7**|
>
> > **[Q4] How might the CDAM approach be adapted to address challenges in recognizing rare or fine-grained classes that are crucial in open-vocabulary segmentation?**
>
> As discussed in [Q2.2], CLIP-based training-free methods, including our CDAM, SCLIP, ClearCLIP, and GEM, are constrained by CLIP's inherent limitations in handling rare or fine-grained classes due to the restricted class diversity and image-text representation in the pre-trained model. However, unlike other methods, CDAM integrates textual information of target classes to guide visual feature extraction through an attention-based process. Our approach leverages augmented text prompts, such as attributes (e.g., color, texture), to better localize these fine-grained or rare targets in the attention map, thereby mitigating these challenges to some extent.

---

> > ### Comment · Reviewer_6UEY · 2024-11-24
> >
> > I thank the authors for their comprehensive responses. While it would have been better to have these explanations in the submitted version I understand that this is not always possible. Nevertheless, adding these details definitely makes the contribution more clear. Based on the responses (also to the other reviewers) I decided to raise my score.

---

> ### Author Response · Authors · 2024-11-27
>
> Thank you for your thoughtful feedback and for taking the time to review our responses. We are pleased that our responses have improved the clarity of our contributions, and we sincerely appreciate the increased score. The detailed responses have been incorporated into the revised paper. Once again, thank you for your valuable and constructive suggestions.

---

### Official Review · Reviewer_ehJ5 · 2024-11-01

**Soundness:** 2
**Presentation:** 3
**Contribution:** 2
**Rating:** 6
**Confidence:** 4

**Summary:**

This paper proposes a Class Distribution-induced Attention Map (CDAM) to enhance the capability of CLIP features in representing different categories for open-vocabulary semantic segmentation. The proposed method can be easily embedded into various approaches to boost their performance. Additionally, the authors introduce an entropy-based background thresholding technique for semantic segmentation inference to facilitate the extraction of foreground classes. The experiments demonstrate the effectiveness of the proposed methods.

**Strengths:**

1. The paper is well-written, with a clear and accessible presentation.
2. The authors conducted comprehensive experiments that effectively highlight the superiority of the proposed method.

**Weaknesses:**

1. The authors claim that CLIP-based prior works yield patch-wise noisy class predictions while having highly correlated class distributions for each object, but this lacks necessary validation. Although an analysis of CLIP and MaskCLIP is provided in the methods section to support this claim, the analysis is not sufficiently general. MaskCLIP is an earlier work, and more recent research in the field may have addressed patch-wise noisy class predictions. Thus, the argument may not be robust, and the paper's novelty compared to related works remains debatable.

2. Jensen-Shannon divergence is a key technique used in this paper. However, there is insufficient discussion on the necessity of using this method and why alternative techniques would be inadequate.

3. The paper lacks discussion of other relevant methods. Methods such as [1], [2], and [3] involve reusing the CLIP [CLS] token and optimizing the feature space, which could enhance CLIP's performance in region recognition. It remains unclear whether these methods could also address the issues raised in this paper.

4. In Table 2, the performance of ClearCLIP is significantly higher than that of SCLIP, as reported in the original ClearCLIP results. However, after incorporating CDAM, SCLIP outperforms ClearCLIP. The reason for this performance discrepancy requires further explanation.

[1] Side Adapter Network for Open-Vocabulary Semantic Segmentation

[2] Learning Mask-aware CLIP Representations for Zero-Shot Segmentation CLIP-Adapted Region-to-Text Learning

[3] AlignZeg: Mitigating Objective Misalignment for Zero-shot Semantic Segmentation

**Questions:**

My primary concern is that the motivation for this work lacks sufficient support, and there is a lack of necessary explanation for some of the techniques used in the proposed method, as outlined in the weaknesses.

---

> ### Author Response · Authors · 2024-11-22
>
> Thank you for sharing your thoughtful comments and feedback.
>
> > **[W1] The authors claim that CLIP-based prior works yield patch-wise noisy class predictions while having highly correlated class distributions for each object, but this lacks necessary validation. Although an analysis of CLIP and MaskCLIP is provided in the methods section to support this claim, the analysis is not sufficiently general. MaskCLIP is an earlier work, and more recent research in the field may have addressed patch-wise noisy class predictions. Thus, the argument may not be robust, and the paper's novelty compared to related works remains debatable.**
>
> Table 1 provides statistical evidence to support our claim over diverse recent baselines, reporting the results of the following experiment: For a given image, one patch $P_{target}$ was randomly selected and then two patches $P_{in}$ and $P_{out}$ were randomly selected from the target class region and the rest of the region, respectively. Then, we measure (1) the probability of {class prediction in $P_{target}$ is correct} and (2) the probability of {distribution similarity between $P_{target}$ and $P_{in}$ < distribution similarity between $P_{target}$ and $P_{out}$}. These results clearly support that our claim is still valid even for more recent CLIP-based prior works such as SCLIP, ClearCLIP and GEM.
>
> In addition, we have conducted an ablation study to validate the effectiveness of our proposed CDAM components over various baselines including recent works such as SCLIP, ClearCLIP, and GEM. Table 2 shows that the addition of our CDAM components consistently improved open-vocabulary semantic segmentation performance without requiring additional training.
>
> These new studies strengthen our claim and demonstrate the robustness and novelty of our approach over diverse baselines. The corresponding updates have been incorporated into the revised paper (Line 211-221 and Section 4.3)
>
> **Table 1:** Accuracy comparison of class predictions and similarity of class distributions with several CLIP-based training-free methods across datasets. Similarity of class distribution is measured using JS divergence.
>
> |Baseline||VOC21|Context60|COCO-Obj|Avg.|
> |-|-|-|-|-|-|
> |MasKCLIP|Class Prediction|56.1 $\pm$ 1.17|38.4 $\pm$ 0.23|27.4 $\pm$ 0.46|43.0|
> ||Sim of Class Dist|70.9 $\pm$ 0.44|73.1 $\pm$ 0.34|69.8 $\pm$ 0.42|71.0|
> |SCLIP|Class Prediction|67.0 $\pm$ 0.49|41.8 $\pm$ 0.31|33.6 $\pm$ 0.23|47.4|
> ||Sim of Class Dist|78.9 $\pm$ 0.26|72.0 $\pm$ 0.30|75.4 $\pm$ 0.55|75.5|
> |ClearCLIP|Class Prediction|70.3 $\pm$ 0.51|42.7 $\pm$ 0.23|36.4 $\pm$ 0.19|49.9|
> ||Sim of Class Dist|76.0 $\pm$ 0.58|70.5 $\pm$ 0.58|71.7 $\pm$ 0.48|72.3|
> |GEM|Class Prediction|70.8 $\pm$ 1.12|42.4 $\pm$ 0.38|37.5 $\pm$ 0.45|50.0|
> ||Sim of Class Dist|79.4 $\pm$ 0.89|71.2 $\pm$ 0.34|74.2 $\pm$ 0.20|74.8|
>
>
> **Table 2:**  Ablation study on components of our CDAM with several baseline methods. We measured the performance on VOC21. ATP refers to the augmented text prompts. Ba denotes Baseline, At-C denotes $\text{Attn} _\text{CDAM}$, At-M denotes $\text{Attn} _\text{MS}$, ATP refers to the augmented text prompts and Th-e denotes $\text{Thr} _\text{ent-bg}$. The evaluation is based on mIoU(\%).
>
> |Ba|At-C|At-M|ATP|Th-e|MaskCLIP|SCLIP|ClearCLIP|GEM|
> |-|-|-|-|-|-|-|-|-|
> |✔| | | | |33.1|50.5|50.7|52.1|
> |✔|✔| |||50.1|55.0|52.1|54.7|
> |✔|✔|✔|||53.7|56.9|55.8|56.5|
> |✔|✔|✔|✔||54.7|57.2|56.0|56.9|
> |✔|✔|✔|✔|✔|**55.9**|**59.0**|**57.6**|**58.7**|
>
> > **[W2] Jensen-Shannon divergence is a key technique used in this paper. However, there is insufficient discussion on the necessity of using this method and why alternative techniques would be inadequate.**
>
> We have conducted an ablation study using several baseline methods with diverse metrics such as Kullback–Leibler (KL) divergence and Wasserstein distance (WS). Table 3 demonstrates that JS divergence consistently achieves the best averaged performance across three benchmarks. We chose JS divergence due to its symmetricity and now Table 3 supports our choice of JS divergence sufficiently. We have updated these results in revised paper (Line 465-468 and Line 890-893).
>
> **Table 3:** Ablation study of similarity metrics for measuring the distance of class distributions over patches. WS refers to the Wasserstein distance. The evaluation is based on mIoU(\%).
>
> |Method|Metric|VOC21|Context60|COCO-Obj|Avg.|
> |-|-|-|-|-|-|
> |MaskCLIP+CDAM|KL|55.7|30.4|34.3|40.1|
> ||JS|55.9|30.5|34.3|**40.2**|
> ||WS|53.0|26.7|28.5|36.1|
> |SCLIP+CDAM|KL|58.8|30.4|34.6|**41.3**|
> ||JS|59.0|30.4|34.5|**41.3**|
> ||WS|57.2|29.0|31.4|39.2|
> |ClearCLIP+CDAM|KL|57.4|29.5|34.3|40.4|
> ||JS|57.6|29.8|34.5|**40.6**|
> ||WS|56.9|28.6|33.4|39.6|
> |GEM+CDAM|KL|58.9|30.5|35.1|**41.5**|
> ||JS|58.7|30.6|35.2|**41.5**|
> ||WS|58.4|29.5|34.0|40.6|

---

> ### Author Response · Authors · 2024-11-22
>
> > **[W3] The paper lacks discussion of other relevant methods. Methods such as [1], [2], and [3] involve reusing the CLIP [CLS] token and optimizing the feature space, which could enhance CLIP's performance in region recognition. It remains unclear whether these methods could also address the issues raised in this paper.**
>
> There are a number of fundamental differences between the aforementioned methods [1, 2, 3] and our proposed CDAM, but one important difference is that the former uses ground truth labels for supervised training, but the latter, our CDAM, is training-free, thus no ground truth needed. The use of ground truth label makes it hard to analyze if the methods of [1, 2, 3] can potentially address the raised issue without ground truth labels just like our CDAM. We have incorporated this discussion into the revised paper under the "Related Works" section to address this point more comprehensively.
>
> > **[W4] In Table 2, the performance of ClearCLIP is significantly higher than that of SCLIP, as reported in the original ClearCLIP results. However, after incorporating CDAM, SCLIP outperforms ClearCLIP. The reason for this performance discrepancy requires further explanation.**
>
> This discrepancy can be explained by the structural differences between SCLIP and ClearCLIP. ClearCLIP removed the residual connection in its architecture to suppress noisy class predictions. Since CDAM uses class distribution similarity to overcome noisy class prediction, the advantage of ClearCLIP had been dimished, thus less performace performance gain was obtained with CDAM.

---

> ### Author Response · Authors · 2024-11-26
>
> Dear Reviewer ehJ5,
>
> We sincerely appreciate your valuable feedback and thoughtful comments. We have carefully addressed your concerns, particularly regarding the sufficient support for our motivation and the explanations of the techniques (JS divergence). As the discussion period is drawing to a close, we would be most grateful if you could kindly let us know whether our responses have sufficiently resolved your concerns. Please let us know if further clarifications are needed.
>
> Thank you again for your time and kind consideration.
>
> Best regards, Authors

---

> > ### Comment · Reviewer_ehJ5 · 2024-11-26
> >
> > The author has addressed most of my concerns, so I will increase the score. Regarding the JS divergence issue, the experimental validation can only prove its effectiveness; the paper should further discuss the underlying reasons for the effectiveness of the method in relation to JS divergence.

---

> ### Author Response · Authors · 2024-11-27
>
> We are pleased to have addressed most of your concerns and the clarifications in our responses have been incorporated into the revised paper. We also sincerely appreciate your suggestion to further clarify our work.
>
> As suggested, in addition to providing experimental validation of the effectiveness of JS divergence, we will include a discussion on the rationale for using JS divergence over others in the construction of our CDAM as below. It seems natural for the similarity of class distributions to have the properties of symmetricity (should be the same regardless of the input order) and permutation invariance (should be the same regardless of class order). The former is satisfied by Wasserstein (WS) distance and JS divergences and the latter is satisfied by KL and JS divergences. Note that it is not straightforward to properly define metrics among classes for the problem of semantic segmentation, which makes WS distance not suitable for our CDAM. Moreover, divergence-based metrics such as KL and JS divergences are known to be more sensitive to small changes over WS distance [1], which may be advantageous for our CDAM. Thus, JS distance seems to have favorable properties that can be used for measuring the distance between class distributions, wihch is consistent with our experimental validation that we have provided before.
>
> [1] Ozair, Sherjil, et al. "Wasserstein dependency measure for representation learning." NeurIPS (2019).

---

### Official Review · Reviewer_874d · 2024-11-02

**Soundness:** 3
**Presentation:** 3
**Contribution:** 3
**Rating:** 8
**Confidence:** 3

**Summary:**

Current CLIP-based Open-Vocabulary Semantic Segmentation (OVSS) methods face limitations due to noisy patch-wise class predictions and highly correlated class distributions for each object. To address these issues, the authors propose a Class Distribution-induced Attention Map (CDAM), generated using the Jensen-Shannon divergence between class distributions of two patches from the same object, to enhance focus on class-relevant regions without additional training. The authors also introduce enhancements such as multi-scale image patches, augmented text prompts, and entropy-based background thresholding to further improve CDAM. Comprehensive experiments demonstrate that CDAM improves multiple OVSS methods across several datasets, with ablation studies validating the effectiveness of each component.

**Strengths:**

1. The motivation is good, with CDAM introducing a novel approach to leverage class distributions for enhancing OVSS performance, offering a new pathway for training-free segmentation improvements.
2. The paper is well-written, with clear explanations of the methodology, experimental settings and results.
3. CDAM is training-free, and the authors provide comprehensive quantitative results demonstrating the effectiveness of each component.

**Weaknesses:**

Writing Suggestions
1. The authors state that “CLIP-based prior works yield patch-wise noisy class predictions while having highly correlated class distributions for each object.” Is this conclusion based on statistical analysis, or is it an observation from limited examples? Providing more statistical evidence would make this argument more convincing.
2. In the third paragraph of Section 1 (lines 51 to 60), the authors should provide additional details on how CDAM is constructed and, more importantly, explain why it is effective. Focusing on why it works would strengthen this section.
3. The order of Figure 1 and Figure 2 should be swapped, as Figure 2 is referenced before Figure 1 (lines 125 to 126).

**Questions:**

1. Similar to Table 3, an additional ablation study on CDAM components using another baseline model would be beneficial.
2. Since CDAM relies on distributions between patches, an ablation study on the impact of different patch sizes would be beneficial.
3. A detailed analysis of the computational complexity of the CDAM would be helpful. Additional experiments related to runtime or efficiency when constructing CDAM would add value.
4. Comparing Table 1 and Table 2, there is a noticeable difference in CDAM’s improvement with/without the background class. The authors should provide more explanation for these differences to clarify their impact on performance.

---

> ### Author Response · Authors · 2024-11-22
>
> We appreciate your constructive comments and suggestions.
>
> > **[W1] The authors state that “CLIP-based prior works yield patch-wise noisy class predictions while having highly correlated class distributions for each object.” Is this conclusion based on statistical analysis, or is it an observation from limited examples? Providing more statistical evidence would make this argument more convincing.**
>
> Table 1 provides statistical evidence to support our claim, reporting the results of the following experiment: For a given image, one patch $P_{target}$ was randomly selected and then two patches $P_{in}$ and $P_{out}$ were randomly selected from the target class region and the rest of the region, respectively. Then, we measure (1) the probability of {class prediction in $P_{target}$ is correct} and (2) the probability of {distribution similarity between $P_{target}$ and $P_{in}$ < distribution similarity between $P_{target}$ and $P_{out}$}. Clearly, our claim "CLIP-based prior works yield patch-wise noisy class predictions while having highly correlated class distributions for each object" can be supported by these results. These results were reflected in the revision (Line 211-221).
>
> **Table 1:** Accuracy comparison of class predictions and similarity of class distributions with several CLIP-based training-free methods across datasets. Similarity of class distribution is measured using JS divergence.
>
> |Baseline||VOC21|Context60|COCO-Obj|Avg.|
> |-|-|-|-|-|-|
> |MasKCLIP|Class Prediction|56.1 $\pm$ 1.17|38.4 $\pm$ 0.23|27.4 $\pm$ 0.46|43.0|
> ||Sim of Class Dist|70.9 $\pm$ 0.44|73.1 $\pm$ 0.34|69.8 $\pm$ 0.42|71.0|
> |SCLIP|Class Prediction|67.0 $\pm$ 0.49|41.8 $\pm$ 0.31|33.6 $\pm$ 0.23|47.4|
> ||Sim of Class Dist|78.9 $\pm$ 0.26|72.0 $\pm$ 0.30|75.4 $\pm$ 0.55|75.5|
> |ClearCLIP|Class Prediction|70.3 $\pm$ 0.51|42.7 $\pm$ 0.23|36.4 $\pm$ 0.19|49.9|
> ||Sim of Class Dist|76.0 $\pm$ 0.58|70.5 $\pm$ 0.58|71.7 $\pm$ 0.48|72.3|
> |GEM|Class Prediction|70.8 $\pm$ 1.12|42.4 $\pm$ 0.38|37.5 $\pm$ 0.45|50.0|
> ||Sim of Class Dist|79.4 $\pm$ 0.89|71.2 $\pm$ 0.34|74.2 $\pm$ 0.20|74.8|
>
> > **[W2] In the third paragraph of Section 1 (lines 51 to 60), the authors should provide additional details on how CDAM is constructed and, more importantly, explain why it is effective. Focusing on why it works would strengthen this section.**
>
> We have revised the third paragraph of Section 1 to emphasize why leveraging the similarity of class distributions is effective for generating attention maps, as implemented in CDAM, for segmentation. The reason why it works is now supported by a new statistical study in the above Table (see the response to [W1]).
>
> > **[W3] The order of Figure 1 and Figure 2 should be swapped, as Figure 2 is referenced before Figure 1 (lines 125 to 126).**
>
> We have swapped the positions of Figure 1 and Figure 2 to align with the reference order.

---

> > ### Author Response · Authors · 2024-11-22
> >
> > > **[Q1] Similar to Table 3, an additional ablation study on CDAM components using another baseline model would be beneficial.**
> >
> > We conducted additional experiments with several alternative baseline models, including MaskCLIP, SCLIP, ClearCLIP, and GEM, as suggested. As shown in Table 2 (below), the proposed CDAM components improve open-vocabulary semantic segmentation performance without requiring additional training. We have updated Table 3 in the main paper and revised Section 4.3 accordingly.
> >
> > **Table 2:**  Ablation study on components of our CDAM with several baseline methods. We measured the performance on VOC21. Ba denotes Baseline, At-C denotes $\text{Attn} _\text{CDAM}$, At-M denotes $\text{Attn} _\text{MS}$, ATP refers to the augmented text prompts and Th-e denotes $\text{Thr} _\text{ent-bg}$. The evaluation is based on mIoU(\%).
> >
> > |Ba|At-C|At-M|ATP|Th-e|MaskCLIP|SCLIP|ClearCLIP|GEM|
> > |-|-|-|-|-|-|-|-|-|
> > |✔| | | | |33.1|50.5|50.7|52.1|
> > |✔|✔| |||50.1|55.0|52.1|54.7|
> > |✔|✔|✔|||53.7|56.9|55.8|56.5|
> > |✔|✔|✔|✔||54.7|57.2|56.0|56.9|
> > |✔|✔|✔|✔|✔|**55.9**|**59.0**|**57.6**|**58.7**|
> >
> > > **[Q2] Since CDAM relies on distributions between patches, an ablation study on the impact of different patch sizes would be beneficial.**
> >
> > To analyze the impact of different patch sizes on our proposed CDAM, we have conducted benchmark experiments using CLIP ViT/B-32 (patch size = 32), reporting the results in Table 3 below. Note that our submission presented the results using CLIP ViT/B-16 (patch size = 16). Other patch size experiment requires a new pre-trained CLIP model, but there was no model with the patch size of 8.
> >
> > Table 3 demonstrates that our proposed CDAM effectively enhances the performance of CLIP-based training-free segmentation methods with a larger patch size. It also shows that large patch size (32) usually resulted in decreased performance as compared to small patch size (16) in Table 1 of the main paper. It seems that small patch size (16) is advantageous over large patch size (32) for segmentation due to better spatial resolution of segmentation. Detailed results are included in the suppl material (Line 1023-1060).
> >
> > **Table 3:**  Ablation study with CLIP ViT/B-32 for exploring different patch sizes. Performance improvements achieved by CDAM are indicated in parentheses. The evaluation metric used is mIoU (%).
> >
> > |Method|VOC21|Context60|COCO-Obj|Avg.|
> > |-|-|-|-|-|
> > |MaskCLIP|29.5|8.1|11.5|16.4|
> > |MaskCLIP+**CDAM**|50.1 (+20.6)|27.6 (+19.5)|27.8 (+16.3)|35.2 (+18.8)|
> > |SCLIP|38.0|24.1|25.1|29.1|
> > |SCLIP+**CDAM**|51.6 (+13.6)|25.7 (+1.6)|27.6 (+2.5)|35.0 (+5.9)|
> > |ClearCLIP|47.6|23.3|27.3|32.7|
> > |ClearCLIP+**CDAM**|51.4 (+3.8)|27.5 (+4.2)|28.4 (+1.1)|35.8 (+3.1)|
> > |GEM|52.1|28.1|33.8|38.0|
> > |GEM+**CDAM**|**55.9** (+3.8)|**32.4** (+4.3)|**34.4** (+0.6)|**40.9** (+2.9)|

---

> ### Author Response · Authors · 2024-11-22
>
> > **[Q3] A detailed analysis of the computational complexity of the CDAM would be helpful. Additional experiments related to runtime or efficiency when constructing CDAM would add value.**
>
> We have conducted two experiments (1 NVIDIA A100 GPU used): (1) a comparison of inference time with several baselines (Table 4) and (2) an analysis of the inference time for each component of our CDAM (Table 5). Considering how much CDAM can improve performance (to the level of CaR!), the additional computation time can be justified and one can still obtain the result for one image within at most 60 msec (0.06 sec), which may still suitable for many real-time applications. Note that Table 5 shows that the majority of the computational cost in CDAM arises from its multi-scale image patches ($\text{Attn} _\text{MS}$) and augmented text prompts (ATP). The computational overhead increases with the number of scales ($m$) for $\text{Attn} _\text{MS}$ and the number of classes for ATP, primarily due to the increased computational burden of computing Jensen-Shannon (JS) divergence. These results have been included in the revised paper (Line 460-464 and Line 782-804).
>
> **Table 4:** Inference time comparison (in seconds per image) of baseline methods with our CDAM.
>
> |Method|VOC21|Context60|COCO-Obj|
> |-|-|-|-|
> |CaR|3.497 sec|9.340 sec|12.270 sec|
> |CLIP-DIY|0.520 sec|-|0.559 sec|
> |MaskCLIP|0.017 sec|0.017 sec|0.017 sec|
> |MaskCLIP+**CDAM**|0.043 sec (+0.026)|0.049 sec (+0.032)|0.051 sec (+0.034)|
> |SCLIP|0.018 sec|0.018 sec|0.018 sec|
> |SCLIP+**CDAM**|0.044 sec (+0.026)|0.050 sec (+0.032)|0.052 sec (+0.034)|
> |ClearCLIP|0.017 sec|0.018 sec|0.018 sec|
> |ClearCLIP+**CDAM**|0.044 sec (+0.027)|0.050 sec (+0.032)|0.051 sec (+0.033)|
> |GEM|0.026 sec|0.026 sec|0.026 sec|
> |GEM+**CDAM**|0.052 sec (+0.026)|0.059 sec (+0.033)|0.060 sec (+0.034)|
>
>
> **Table 5:** Inference time (in seconds per image) for each component of CDAM. The baseline model is set as MaskCLIP, as the additional overhead introduced by CDAM is consistent across other baseline methods. Ba denotes Baseline, At-C denotes $\text{Attn} _\text{CDAM}$, At-M denotes $\text{Attn} _\text{MS}$, ATP refers to the augmented text prompts and Th-e denotes $\text{Thr} _\text{ent-bg}$.
>
> |Ba|At-C|At-M|ATP|Th-e|VOC21|Context60|COCO-Obj
> |-|-|-|-|-|-|-|-|
> |✔| | | | |0.017 sec|0.017 sec|0.017 sec|
> |✔|✔| |||0.020 sec|0.022 sec|0.024 sec|
> |✔|✔|✔|||0.032 sec|0.038 sec|0.040 sec|
> |✔|✔|✔|✔||0.043 sec|0.049 sec|0.051 sec|
> |✔|✔|✔|✔|✔|0.043 sec|0.049 sec|0.051 sec|
>
> > **[Q4] Comparing Table 1 and Table 2, there is a noticeable difference in CDAM’s improvement with/without the background class. The authors should provide more explanation for these differences to clarify their impact on performance.**
>
> These differences stem from the different datasets and their corresponding evaluation metrics. Table 1 used the datasets with "background" class so that the background must be predicted accurately for better performance. However, Table 2 used the datasets where no background class is defined so that the accurate background estimation does not take into account for performance evaluation. Thus, the observed differences in CDAM's performance improvement between datasets (Table 1 vs. Table 2) are primarily attributed to the evaluation results with and without background class. Specifically, in Table 2 (no background class), inaccurate predictions in background areas have minimal influence on the evaluation metric, reducing their impact on the reported performance. Thus, CDAM's advantage of predicting background accurately with reduced false positive led to significant performance improvement in Table 1, but smaller performance gain in Table 2. This explanation has been included in the revised paper (Line 414-417).

---

> > ### Comment · Reviewer_874d · 2024-11-24
> > **Thanks**
> >
> > Thank you for your detailed responses and clarifications. I greatly appreciate your efforts. Kindly incorporate the modified content into the main paper where possible. If this is not possible, please add them to the supplementary material. These additions will significantly enhance the paper’s readability. Based on these changes, I will increase my rating.

---

> ### Author Response · Authors · 2024-11-27
>
> Thank you for your detailed suggestions for improving the clarity of our work. We also appreciate the increase in your score. We have incorporated our responses into both the main paper and the supplementary material for better understanding and enhanced readability. Thank you once again for your valuable feedback.

---

### Official Review · Reviewer_cFei · 2024-11-05

**Soundness:** 3
**Presentation:** 4
**Contribution:** 3
**Rating:** 8
**Confidence:** 4

**Summary:**

The noisy patch-level prediction is rectified by class distribution-induced attention in zero-shot open-vocabulary semantic segmentation in this paper. The motivation is clear. Experimental results show the effectiveness of the proposed idea cooperating with several state-of-the-art approaches and significant performance improvement. The paper is well-written and the figures are easy to follow.

**Strengths:**

1. The paper is good enough for me as the clear motivation and consistent performance gain on several public semantic segmentation datasets when integrating the proposed Class Distribution-induced Attention Map to different SOTA methods.
2. The motivation is clear and illustrated well in Fig. 1.

**Weaknesses:**

1. In Table 1, even though CaR is a heavily computational method and CLIP-DIY uses an extra background extractor,  the proposed CDAM is not integrated into CaR and CLIP-DIY, and the best performance is not achieved on VOC21.

**Questions:**

1. It's better to give the mIoU both for seen and unseen classes.

---

> ### Author Response · Authors · 2024-11-22
>
> We sincerely appreciate your thoughtful review and valuable feedback.
>
> > **[W1] In Table 1, even though CaR is a heavily computational method and CLIP-DIY uses an extra background extractor, the proposed CDAM is not integrated into CaR and CLIP-DIY, and the best performance is not achieved on VOC21.**
>
> Many recent CLIP-based training-free classification methods rely on task-agnostic local visual tokens and our CDAM can be seamlessly integrated with them for improved performance. However, CaR and CLIP-DIY use task-specific CLS tokens for region classification, so it is not straightforward to incorporate them with our CDAM yet.
>
> Note that our CDAM with GEM outperformed all prior works including CaR and CLIP-DIY on average over all benchmark datasets as well as on both Context60 and COCO-Obj. Our CDAM with SCLIP yielded the second best performance (comparable to the first!) on VOC21, but with incredibly fast inference time (CaR 59.4% with 3.497 sec per image vs. SCLIP+CDAM 59.0% with 0.043 sec per image). See the below table for inference time.
>
> Table 1: Inference time comparison (seconds per image).
>
> |Method|VOC21|Context60|COCO-Obj|
> |-|-|-|-|
> |CaR|3.497 sec|9.340 sec|12.270 sec|
> |CLIP-DIY|0.520 sec|-|0.559 sec|
> |MaskCLIP|0.017 sec|0.017 sec|0.017 sec|
> |MaskCLIP+**CDAM**|0.043 sec (+0.026)|0.049 sec (+0.032)|0.051 sec (+0.034)|
> |SCLIP|0.018 sec|0.018 sec|0.018 sec|
> |SCLIP+**CDAM**|0.044 sec (+0.026)|0.050 sec (+0.032)|0.052 sec (+0.034)|
> |ClearCLIP|0.017 sec|0.018 sec|0.018 sec|
> |ClearCLIP+**CDAM**|0.044 sec (+0.027)|0.050 sec (+0.032)|0.051 sec (+0.033)|
> |GEM|0.026 sec|0.026 sec|0.026 sec|
> |GEM+**CDAM**|0.052 sec (+0.026)|0.059 sec (+0.033)|0.060 sec (+0.034)|
>
> > **[Q1] It's better to give the mIoU both for seen and unseen classes.**
>
> As you suggested, open-vocabulary semantic segmentation methods can be evaluated either for seen / unseen classes or unseen classes only. However, we have focused on unseen classes only since our method is training-free and all 16 prior works in Table 1 including CaR, CLIP-DIY, MaskCLIP, SCLIP, ClearCLIP, GEM were also evaluated for unseen classes only. Extending this work for the case with seen / unseen classes, however, can be a great future work.

---

> > ### Comment · Reviewer_cFei · 2024-11-28
> >
> > I'm glad to see other reviewers raise their scores as I really like simple yet effective methods. For the performance of unseen classes, it just needs to run the evaluation code very quickly. I don't know why it cannot provided. I doubt whether the method was not good enough in seen classes. If a model worked well on unseen classes and it forgot the seen classes, I think it is not practical.

---

> ### Author Response · Authors · 2024-12-02
>
> We appreciate your recognition on our method's simplicity and effectiveness.
>
> It seems that your question may stem from some existing works such as [1,2] that used mask annotations for specific classes in supervised ways (e.g., prompt tuning and/or tuning additional decoder) while their titles contain "zero-shot" to emphasize their capability of deal with unseen classes as well. These methods, thus, can be evaluated for both seen and unseen classes. See CaR[3] and SegCLIP[4] for the clear differences between mask annotation-free and annotation-based methods. While our work has focused on annotation-free methods (see below for details), we also share your concern on the potential catastrophic forgetting for seen classes. Thus, we performed additional experiments for mask annotation-based supervision methods with our CDAM and the results are reported in Table 2. As demonstrated, our CDAM did not degrade the performance for seen classes that were optimized by the training-based methods with mask annotation (ZegCLIP[1], OTSeg[2]) while consistently improving the performance for unseen classes.
>
> Table 2: The results of semantic segmentation methods supervised with segmentation mask supervision [1,2] + CDAM on VOC 2012. hIoU refers to harmonic mean IoU.
>
> |Method|mIoU (Seen)| mIoU (Unseen) | hIoU|
> |-|-|-|-|
> |ZegCLIP[1]| 91.8|77.9| 84.3|
> |ZegCLIP[1] + CDAM | 91.8 (+0.0)| 78.4 (+0.5)|84.6 (+0.3)|
> |OTSeg[2]| 93.3|81.8|87.2|
> |OTSeg[2] + CDAM | 93.4 (+0.1)|82.2 (+0.4)|87.4 (+0.2)|
>
> As you may know, if a model is trained or tuned with mask annotations for some classes, those classes are called "seen classes." In contrast, the methods like our proposed method and 16 compared prior arts in Table 1 of the main paper (e.g., SCLIP, ClearCLIP, SegCLIP, FreeDA, etc.) do NOT use any mask annotation. To clarify further, there are three distinct categories for the methods: (1) **mask annotation-based** supervision methods (e.g., ZegCLIP, OTSeg), thus having "seen classes" (2) **mask annotation-free, weakly-supervised** methods that leverage image-text paired datasets (e.g., GroupViT, SegCLIP, etc.), thus NOT having "seen classes" and (3) **mask annotation-free**, training-free methods, including our method (e.g., SCLIP, ClearCLIP, CaR, etc.), thus clearly NOT having "seen classes". Since all methods in Table 1 (Category 2 and 3) were neither trained nor tuned with any mask annotation, there is no seen class. Thus, the evaluations of ours as well as these other prior works have focused exclusively on unseen classes. This has been clarified in the "Related Works" section of the revised paper.
>
> [1] Zhou, Ziqin, et al. "Zegclip: Towards adapting clip for zero-shot semantic segmentation." CVPR (2023).
>
> [2] Kim, Kwanyoung, et al. "OTSeg: Multi-prompt Sinkhorn Attention for Zero-Shot Semantic Segmentation." ECCV (2024).
>
> [3] Sun, Shuyang, et al. "Clip as rnn: Segment countless visual concepts without training endeavor." CVPR (2024).
>
> [4] Luo, Huaishao, et al. "Segclip: Patch aggregation with learnable centers for open-vocabulary semantic segmentation." PMLR (2023).

---

### Meta-Review · Area_Chair_bW5x · 2024-12-23

**Metareview:**

This paper tackles the problem of open-vocabulary semantic segmentation, aiming to address the limitations due to noisy patch-wise class predictions in existing CLIP-based approaches. To this end, the authors propose the class-distribution-induced attention map, generated using the Jensen-Shannon divergence between class distributions of two patches from the same object, to enhance focus on class-relevant regions without additional training. They also introduce enhancements such as multi-scale image patches, augmented text prompts, and entropy-based background thresholding to improve the performance further. Experimental results show the effectiveness of the proposed idea, which cooperates with several state-of-the-art approaches and results in significant performance improvement.
All reviewers appreciated the clear motivation, reasonable idea, and comprehensive analysis/experiments. The main concerns raised by reviewers were limited novelty, unclear exposition, and missing discussion/comparisons. The authors’ detailed rebuttal addressed most of them, resulting in unanimous acceptance at the end of the discussion. AC thus recommends acceptance.

**Additional Comments On Reviewer Discussion:**

The main concerns raised by reviewers were limited novelty, unclear exposition, and missing discussion/comparisons. The authors’ rebuttal addressed most of the concerns so that all reviewers either remained positive or raised their scores after discussion. AC finds no significant issues remained for publication.

---

### Decision · Program_Chairs · 2025-01-22

Accept (Poster)